# Identifying prostate cancer and its clinical risk in asymptomatic men using machine learning of high dimensional peripheral blood flow cytometric natural killer cell subset phenotyping data

Simon P Hood[1†‡], Georgina Cosma[2†*], Gemma A Foulds[1,3], Catherine Johnson[1,3], Stephen Reeder[1,3], Stéphanie E McArdle[1,3], Masood A Khan[4], A Graham Pockley[1,3†*]

[1]John van Geest Cancer Research Centre, School of Science and Technology, Nottingham Trent University, Nottingham, United Kingdom; [2]Department of Computer Science, Loughborough University, Loughborough, United Kingdom; [3]Centre for Health, Ageing and Understanding Disease (CHAUD), School of Science and Technology, Nottingham Trent University, Nottingham, United Kingdom; [4]Department of Urology, University Hospitals of Leicester NHS Trust, Leicester, United Kingdom

*For correspondence:
g.cosma@lboro.ac.uk (GC);
graham.pockley@ntu.ac.uk (AGP)

[†]These authors contributed equally to this work

Present address: [‡]Cancer Research UK Manchester Institute, University of Manchester, Manchester, United Kingdom

**Abstract** We demonstrate that prostate cancer can be identified by flow cytometric profiling of blood immune cell subsets. Herein, we profiled natural killer (NK) cell subsets in the blood of 72 asymptomatic men with Prostate-Specific Antigen (PSA) levels < 20 ng ml$^{-1}$, of whom 31 had benign disease (no cancer) and 41 had prostate cancer. Statistical and computational methods identified a panel of eight phenotypic features ($CD56^{dim}CD16^{high}$, $CD56^+DNAM-1^-$, $CD56^+LAIR-1^+$, $CD56^+LAIR-1^-$, $CD56^{bright}CD8^+$, $CD56^+NKp30^+$, $CD56^+NKp30^-$, $CD56^+NKp46^+$) that, when incorporated into an Ensemble machine learning prediction model, distinguished between the presence of benign prostate disease and prostate cancer. The machine learning model was then adapted to predict the D'Amico Risk Classification using data from 54 patients with prostate cancer and was shown to accurately differentiate between the presence of low-/intermediate-risk disease and high-risk disease without the need for additional clinical data. This simple blood test has the potential to transform prostate cancer diagnostics.

## Introduction

Early diagnosis and treatment increase curative rates for many cancers. The WHO considers that the burden of cancer on health services can be reduced by early detection and that this is achievable via three integrated steps: 1) awareness and accessing care, 2) clinical evaluation, diagnosis, and staging, 3) access to treatment (http://www.who.int/mediacentre/factsheets/fs297/en/). Although the clinical introduction of the Prostate-Specific Antigen (PSA) test in 1986 increased the early diagnosis of localized prostate cancer (*Catalona et al., 1991*; *Hankey et al., 1999*), elevated PSA levels are not necessarily indicative of prostate cancer because PSA levels can be raised by prostatitis, other localised infections, benign hyperplasia and/or factors such as physical stress. Contrastingly, 15% of men with 'normal' PSA levels typically have prostate cancer, with a further 15% of these cancers being high-grade (https://prostatecanceruk.org/prostate-information/prostate-tests/psa-test). The

**eLife digest** With an estimated 1.8 million new cases in 2018 alone, prostate cancer is the fourth most common cancer in the world. Catching the disease early increases the chances of survival, but this cancer remains difficult to detect.

The best diagnostic test currently available measures the blood level of a protein called the prostate-specific antigen (PSA for short). Heightened amounts of PSA may mean that the patient has cancer, but 15% of individuals with prostate cancer have normal levels of the protein, and many healthy people can have high amounts of PSA. This blood test is therefore not widely accepted as a reliable diagnostic tool.

Other methods exist to detect prostate cancer, yet their results are limited. A small piece of the prostate can be taken for analysis, but results from this invasive procedure are often incorrect. Scans can help to spot a tumor, but they are not accurate enough to be conclusive on their own. New tests are therefore urgently needed.

Prostate cancer is often associated with changes in the immune system that can be detected through a blood test. In particular, the appearance of a type of white blood (immune) cells called natural killer cells may be altered. Yet, it was unclear whether measurements based on these cells could help to detect prostate cancer and assess the severity of the disease.

Here, Hood, Cosma et al. collected and examined the natural killer cells of 72 participants with slightly elevated PSA levels and no other symptoms. Amongst these, 31 individuals had prostate cancer and 41 were healthy. These biological data were then used to produce computer models that could detect the presence of the disease, as well as assess its severity. The algorithms were developed using machine learning, where previous patient information is used to make prediction on new data. This work resulted in a new detection tool which was 12.5% more accurate than the PSA test in detecting prostate cancer; and in a detection tool that was 99% accurate in predicting the risk of the disease (in terms of clinical significance) in individuals with prostate cancer.

Although these new approaches first need to be validated in the clinic before being deployed, they could ultimately improve the detection and diagnosis of prostate cancer, saving lives and reducing the need for further tests.

reliable diagnosis of prostate cancer based on PSA levels alone is therefore not possible and confirmation using invasive biopsies is currently required. In 2011/12 approximately 32,000 diagnostic biopsies (28,000 TRUS and 4,000 TPTPB) were performed by the NHS in England (*NICE, 2014*). Although the transrectal ultrasound guided prostate (TRUS) biopsy is the most commonly used technique, it is limited to taking 10 to 12 biopsies primarily from the peripheral zone of the prostate and has a positive detection rate between 26% and 33% (*Aganovic et al., 2011*; *Nafie et al., 2014a*; *Naughton et al., 2000*; *Yuasa et al., 2008*). The Transperineal Template Prostate biopsy (TPTPB) is a 36 core technique that samples all regions of the prostate and delivers a better positive detection rate between 55% and 68% (*Dimmen et al., 2012*; *Nafie et al., 2014b*; *Pal et al., 2012*). However, invasive biopsies are painful and associated with a significant risk of potentially serious side-effects such as urosepsis and erectile dysfunction (*Chang et al., 2013*). Given the potential challenges of invasive tests and the risk of significant side-effects, considerable interest in the potential of non-invasive blood or urine-based tests/approaches ('liquid biopsies') for diagnosing disease has developed (*Quandt et al., 2017*). Liquid biopsies can provide information about both the tumour (e.g. circulating cells, cell-free and exosomal DNA and RNA) and the immune response (e.g. immune cell composition and their gene, protein, and exosome expression profiles). Liquid biopsies are minimally invasive and enable serial assessments and 'live' monitoring speedily and cost-effectively (*Quandt et al., 2017*).

Based on the reciprocal interaction between cancer and the immune system, we have proposed that immunological signatures within the peripheral blood (the peripheral blood 'immunome') can discriminate between men with benign prostate disease and those with prostate cancer and thereby reduce the dependency of diagnosis on invasive biopsies. To this end, we have previously shown that the incorporation of a peripheral blood immune phenotyping-based feature set comprising five phenotypic features $CD8^+CD45RA^-CD27^-CD28^-$ ($CD8^+$ Effector Memory cells),

$CD4^+CD45RA^-CD27^-CD28^-$ ($CD4^+$ Effector Memory cells), $CD4^+CD45RA^+CD27^-CD28^-$ ($CD4^+$ Terminally Differentiated Effector Memory Cells re-expressing CD45RA), $CD3^-CD19^+$ (B cells), $CD3^+CD56^+CD8^+CD4^+$ (NKT cells) into a computation-based prediction tool enables the better detection of prostate cancer and strengthens the accuracy of the PSA test in asymptomatic men having PSA levels < 20 ng/ml (*Cosma et al., 2017*). Herein, we have extended this new approach to determine if phenotypic profiling of peripheral blood natural killer (NK) cell subsets can also discriminate between the presence of benign prostate disease and prostate cancer in the same cohort of asymptomatic men. We also investigate the potential of the peripheral blood dataset to discriminate between low- or intermediate-risk prostate cancer and high-risk prostate cancer in those men having prostate cancer.

## Results

### Distinguishing between benign prostate disease and prostate cancer: statistical analysis of NK cell phenotypic features and PSA levels

Herein, we consider a 'feature' to be a single phenotypic variable (as determined using flow cytometry) or a pre-grouped set of phenotypic variables, as shown in *Table 1*. It was not possible to discriminate between men with benign prostate disease and men with prostate cancer based on differences between phenotypic features/profiles due to their similarity (*Table 1*, *Figure 1*, *Figure 2*).

These findings highlight the difficulty in identifying combinations of features that can best identify the presence of cancer. These difficulties are compounded by the challenge of identifying the best combination of predictors which comprise $n$ number of features, and that features within a combination, ideally, should not correlate. It is important to evaluate correlations between features, because if two features are highly correlated, then only one of these could serve as a candidate predictor. However, there may be occasions where both features are needed and besides the impact of this on the dimensionality of the dataset, there is no other negative impact. Furthermore, when two features are highly correlated and are important, it may be difficult to decide which feature to remove. *Figure 3* shows the correlations between features, where +1.0 indicates a strong positive correlation between two features, and −1.0 indicates a strong negative correlation between two features.

The Kolmogorov-Smirnov and Shapiro-Wilk tests of normality were carried out to determine whether the dataset is normally distributed, as this would determine the choice of statistical tests, that is whether to use parametric (for normally distributed datasets), or non-parametric (for not normally distributed datasets) tests. The results of the normality tests are shown in *Table 2*. The results revealed that only 7–8 features (depending on the normality test) were normally distributed (with $p > 0.05$), and for the remaining features the p value was less than 0.05 ($p < 0.05$) which indicates that there is a statistically significant difference between the distribution of the data of those features and the normal distribution. Based on the results of the test, we can conclude that the dataset is not normally distributed.

Given that most features in the dataset are not normally distributed, the Kruskal-Wallis (also called the 'one-way ANOVA on ranks', a rank-based non-parametric test) tests were used to check for statistically significant differences between the mean ranks of the NK cell phenotypic features in men with benign prostate disease and patients with prostate cancer rather than its parametric equivalent (one-way analysis of variance, ANOVA). Although the Kruskal-Wallis test did not return any significant differences in the mean PSA values between individuals with benign disease and those with prostate cancer ($\chi^2 = 0$; p=0.949, *Figure 4*), statistically significant differences at the alpha level of $\alpha = 0.05$ in the mean ranks of the $CD56^{bright}CD8^+$ (ID14, p=0.007), $CD56^+NKp30^+$ (ID15, p=0.008), $CD56^+NKp30^-$ (ID16, p=0.031), $CD56^+NKp46^+$ (ID17, p=0.023) populations in men with benign prostate disease and those with prostate cancer (*Table 3*) were observed.

This initial analysis provided insight into which phenotypic features might be good candidates for distinguishing between the presence of benign disease and prostate cancer. The next step was to examine whether using these as inputs into a machine learning algorithm can achieve this. An Ensemble Subspace kNN classifier was developed for the task at hand. The section which follows explains the approaches that were used to compare the diagnostic accuracy of the classifier when using the subset of features derived from the statistical analysis, and those features which were selected as a combination using the Genetic Algorithm (GA) for feature selection.

**Table 1.** Descriptive statistics of the dataset.

| | | Min. | | Max. | | Mean | | Std. | | IQR | | Range | | Diff. |
|---|---|---|---|---|---|---|---|---|---|---|---|---|---|---|
| | | Beni. | Canc. | Beni. | Canc. | Beni. | Canc. | Beni. | Canc. | Beni. | Canc. | Beni. | Canc. | |
| | PSA | 4.70 | 4.70 | 19.00 | 19.00 | 8.26 | 8.34 | 3.31 | 3.28 | 3.30 | 4.08 | 14.30 | 14.30 | −0.08 |
| $CD56^{dim}$ % | | | | | | | | | | | | | | |
| 1 | $CD16^+$ | 83.85 | 73.04 | 96.61 | 96.98 | 90.98 | 90.64 | 3.35 | 5.46 | 4.13 | 5.02 | 12.76 | 23.94 | 0.34 |
| 2 | $CD16^{high}$ | 24.38 | 49.66 | 87.46 | 89.33 | 72.88 | 73.32 | 11.74 | 10.22 | 15.00 | 10.45 | 63.08 | 39.67 | −0.44 |
| 3 | $CD16^{low}$ | 5.17 | 6.57 | 64.22 | 44.00 | 17.74 | 16.84 | 10.40 | 7.45 | 8.76 | 7.66 | 59.05 | 37.43 | 0.90 |
| 4 | $CD16^-$ | 1.41 | 1.25 | 11.11 | 18.06 | 4.83 | 4.89 | 2.45 | 3.48 | 2.58 | 2.68 | 9.70 | 16.81 | −0.06 |
| 5 | $CD56^{dim}total$ | 91.29 | 87.24 | 98.70 | 98.70 | 95.81 | 95.53 | 2.02 | 2.58 | 2.96 | 3.02 | 7.41 | 11.46 | 0.28 |
| $CD56^{bright}$ % | | | | | | | | | | | | | | |
| 6 | $CD16^+$ | 0.46 | 0.65 | 5.10 | 5.88 | 1.91 | 1.83 | 1.06 | 1.04 | 1.64 | 0.92 | 4.64 | 5.23 | 0.08 |
| 7 | $CD16^{high}$ | 0.09 | 0.12 | 1.97 | 1.15 | 0.60 | 0.47 | 0.44 | 0.25 | 0.50 | 0.40 | 1.88 | 1.03 | 0.13 |
| 8 | $CD16^{low}$ | 0.34 | 0.40 | 3.11 | 4.95 | 1.27 | 1.35 | 0.72 | 0.86 | 0.97 | 0.63 | 2.77 | 4.55 | −0.07 |
| 9 | $CD16^-$ | 0.61 | 0.58 | 5.78 | 9.09 | 2.28 | 2.64 | 1.14 | 1.82 | 1.42 | 1.75 | 5.17 | 8.51 | −0.36 |
| 10 | $CD56^{bright}total$ | 1.30 | 1.30 | 8.71 | 12.76 | 4.19 | 4.47 | 2.02 | 2.58 | 2.95 | 3.01 | 7.41 | 11.46 | −0.28 |
| CD8% | | | | | | | | | | | | | | |
| 11 | $CD56^+CD8^+$ | 21.88 | 9.20 | 86.70 | 80.47 | 46.43 | 40.71 | 15.64 | 14.66 | 24.03 | 20.05 | 64.82 | 71.27 | 5.72 |
| 12 | $CD56^+CD8^-$ | 13.30 | 19.53 | 78.12 | 90.80 | 53.57 | 59.29 | 15.64 | 14.66 | 24.03 | 20.05 | 64.82 | 71.27 | −5.72 |
| 13 | $CD56^{dim}CD8^+$ | 19.63 | 8.60 | 82.38 | 77.47 | 45.18 | 39.11 | 15.31 | 14.10 | 24.72 | 19.36 | 62.75 | 68.87 | 6.07 |
| 14 | $CD56^{bright}CD8^+$ | 0.37 | 0.25 | 4.75 | 6.64 | 1.41 | 1.70 | 1.07 | 1.41 | 0.70 | 1.60 | 4.38 | 6.39 | −0.29 |
| NKp30 % | | | | | | | | | | | | | | |
| 15 | $CD56^+NKp30^+$ | 40.69 | 56.80 | 96.74 | 98.43 | 79.78 | 88.56 | 16.42 | 10.41 | 21.80 | 10.44 | 56.05 | 41.63 | −8.78 |
| 16 | $CD56^+NKp30^-$ | 3.26 | 1.57 | 58.34 | 44.59 | 20.05 | 11.43 | 16.22 | 10.46 | 20.54 | 10.49 | 55.08 | 43.02 | 8.61 |
| NKp46 % | | | | | | | | | | | | | | |
| 17 | $CD56^+NKp46^+$ | 38.11 | 45.37 | 86.52 | 95.82 | 62.65 | 69.82 | 13.49 | 11.58 | 23.90 | 12.71 | 48.41 | 50.45 | −7.18 |
| 18 | $CD56^+NKp46^-$ | 14.02 | 4.32 | 62.97 | 55.68 | 38.40 | 30.87 | 13.58 | 11.64 | 24.89 | 13.44 | 48.95 | 51.36 | 7.53 |
| DNAM-1 % | | | | | | | | | | | | | | |
| 19 | $CD56^+DNAM-1^+$ | 63.69 | 88.56 | 99.18 | 99.60 | 95.35 | 96.46 | 6.81 | 2.59 | 3.37 | 3.49 | 35.49 | 11.04 | −1.11 |
| 20 | $CD56^+DNAM-1^-$ | 0.86 | 0.42 | 37.29 | 11.66 | 4.74 | 3.59 | 6.96 | 2.61 | 3.45 | 3.54 | 36.43 | 11.24 | 1.14 |
| NKG2D % | | | | | | | | | | | | | | |
| 21 | $CD56^+NKG2D^+$ | 85.17 | 80.79 | 98.77 | 98.96 | 93.49 | 94.07 | 4.45 | 4.87 | 6.81 | 3.83 | 13.60 | 18.17 | −0.58 |
| 22 | $CD56^+NKG2D^-$ | 1.22 | 1.03 | 14.76 | 19.12 | 6.44 | 5.84 | 4.36 | 4.76 | 6.80 | 3.96 | 13.54 | 18.09 | 0.60 |
| | PSA | 4.70 | 4.70 | 19.00 | 19.00 | 8.26 | 8.34 | 3.31 | 3.28 | 3.30 | 4.08 | 14.30 | 14.30 | −0.08 |
| NKp44 % | | | | | | | | | | | | | | |
| 23 | $CD56^+NKp44^+$ | 0.43 | 0.28 | 3.71 | 6.77 | 1.16 | 1.34 | 0.82 | 1.20 | 0.78 | 1.25 | 3.28 | 6.49 | −0.18 |
| 24 | $CD56^+NKp44^-$ | 96.10 | 93.70 | 99.53 | 99.70 | 98.82 | 98.64 | 0.83 | 1.13 | 0.80 | 1.25 | 3.43 | 6.00 | 0.18 |
| CD85j % | | | | | | | | | | | | | | |
| 25 | $CD56^+CD85j^+$ | 19.53 | 14.21 | 84.73 | 91.59 | 53.37 | 55.10 | 19.04 | 18.34 | 30.49 | 20.23 | 65.20 | 77.38 | −1.74 |
| 26 | $CD56^+CD85j^-$ | 14.93 | 8.50 | 81.54 | 86.08 | 46.94 | 45.24 | 19.21 | 18.43 | 30.28 | 21.48 | 66.61 | 77.58 | 1.69 |
| LAIR-1 % | | | | | | | | | | | | | | |
| 27 | $CD56^+LAIR-1^+$ | 94.97 | 21.43 | 99.90 | 99.89 | 99.07 | 97.47 | 1.07 | 12.19 | 0.49 | 0.47 | 4.93 | 78.46 | 1.60 |
| 28 | $CD56^+LAIR-1^-$ | 0.02 | 0.05 | 5.24 | 78.20 | 0.76 | 2.40 | 1.02 | 12.15 | 0.42 | 0.43 | 5.22 | 78.15 | −1.65 |
| NKG2A % | | | | | | | | | | | | | | |
| 29 | $CD56^+NKG2A^+$ | 20.43 | 19.01 | 77.57 | 73.01 | 46.14 | 44.24 | 17.41 | 13.73 | 30.82 | 17.47 | 57.14 | 54.00 | 1.90 |
| 30 | $CD56^+NKG2A^-$ | 22.62 | 27.11 | 79.40 | 80.85 | 54.01 | 55.99 | 17.39 | 13.67 | 30.48 | 17.90 | 56.78 | 53.74 | −1.98 |

*Table 1 continued on next page*

Table 1 continued

| | | Min. | | Max. | | Mean | | Std. | | IQR | | Range | | Diff. |
|---|---|---|---|---|---|---|---|---|---|---|---|---|---|---|
| | | Beni. | Canc. | Beni. | Canc. | Beni. | Canc. | Beni. | Canc. | Beni. | Canc. | Beni. | Canc. | |
| 2B4 % | | | | | | | | | | | | | | |
| 31 | $CD56^+2B4^+$ | 98.41 | 97.06 | 99.99 | 99.96 | 99.53 | 99.50 | 0.39 | 0.59 | 0.32 | 0.33 | 1.58 | 2.90 | 0.02 |
| 32 | $CD56^+2B4^-$ | 0.01 | 0.05 | 1.59 | 2.95 | 0.48 | 0.50 | 0.39 | 0.59 | 0.31 | 0.34 | 1.58 | 2.90 | −0.02 |

Min. is the minimum value, Max. is maximum value, Mean is the mean or average value, and Std. is Standard Deviation. Range is the difference between the minimum and maximum values. The Interquartile range (IQR) is a measure of data variability and was derived by computing the distance between the Upper Quartile (i.e. top) and Lower Quartile (i.e. bottom) of the boxes illustrated in **Figure 1**. Difference is computed as diff = mean(Benign)-mean(Cancer).

## Distinguishing between benign prostate disease and prostate cancer: GA

The GA was used to identify a subset of features that, as a combination, provide an NK cell-based immunophenotypic 'fingerprint' which can determine if an asymptomatic individual with PSA levels below 20 ng ml$^{-1}$ has benign prostate disease or prostate cancer. This fingerprint, or feature set, would then be used to construct a diagnostic/prediction model. Given that GAs stochastically select multiple individuals (i.e. features) from the current population (based on their 'fitness'), each run can return different results. A common approach to identifying the best solution(s) is, therefore, to run the algorithm several times to obtain the frequency of the solution(s). Since the aim herein is to identify the most commonly occurring subset of NK cell phenotypic predictors, the GA was applied to the dataset and the most frequent subset of features returned was considered as being the best and most promising.

Let $f_c$ denote the number of times (frequency) a combination was returned during the $n$ number of runs, then the relative frequency of a combination ($R_{fc}$) can be calculated using formula (**Equation 1**),

$$R_{fc} = \frac{f_c}{n} \tag{1}$$

**Table 4** shows the most frequent feature combinations returned at the end of each of the 30 runs when setting $\lambda$ to different values. In **Table 4**, $\lambda$ is the number of features in a combination. *No. different comb* is the number of unique combinations returned during the $n$ number of runs (i.e. n = 30) for a given $\lambda$; *Comb. with highest freq* is the combination which was returned most frequently during the $n$ number of runs; *Freq of Comb.* is the frequency of the most common combination found in the previous column; *Relative Freq. (%)* is computed using formula (**Equation 1**) converted to a percentage.

As the optimum number of features is not known, the GA was run by setting $\lambda = 2, 3, \ldots, n$ where $n$ is the total number of features in the dataset. **Table 4** shows the results for the first 10 combinations. The results indicate that the combination comprising four features is the most promising in terms of its ability to discriminate between benign prostate disease and prostate cancer on NK cell phenotypic data alone. Features 2, 20, 27, 28, were returned in all 30 runs when searching for the best combination comprising of four features. Furthermore, features 20, 27, 28 were returned together in all combinations comprising more than three features (see feature ID's in combinations $\lambda = 4$ to $\lambda = 10$ in **Table 4**). These results strongly suggest that these are good predictors when grouped. The fact that the same combination was returned in 30 iterations is a strong indicator that these four features are the most reliable for distinguishing between the presence of benign prostate disease and prostate cancer. Although the statistical analysis presented in **Table 3** determined that features: ID14: $CD56^{bright}CD8^+$, ID15: $CD56^+NKp30^+$, ID16: $CD56^+NKp30^-$, and ID17: $CD56^+NKp46^+$ were the only ones with values which were significantly different in the two groups at $\alpha = 0.05$, and for which p values were therefore less than 0.05, none of the features selected by the statistical analysis were returned by the GA when searching for the best combination of features for discriminating between the presence of benign prostate disease and prostate cancer. The features selected by the GA were: ID2: $CD56^{dim}CD16^{high}$, ID20: $CD56^+DNAM-1^-$, ID27: $CD56^+LAIR-1^+$, and

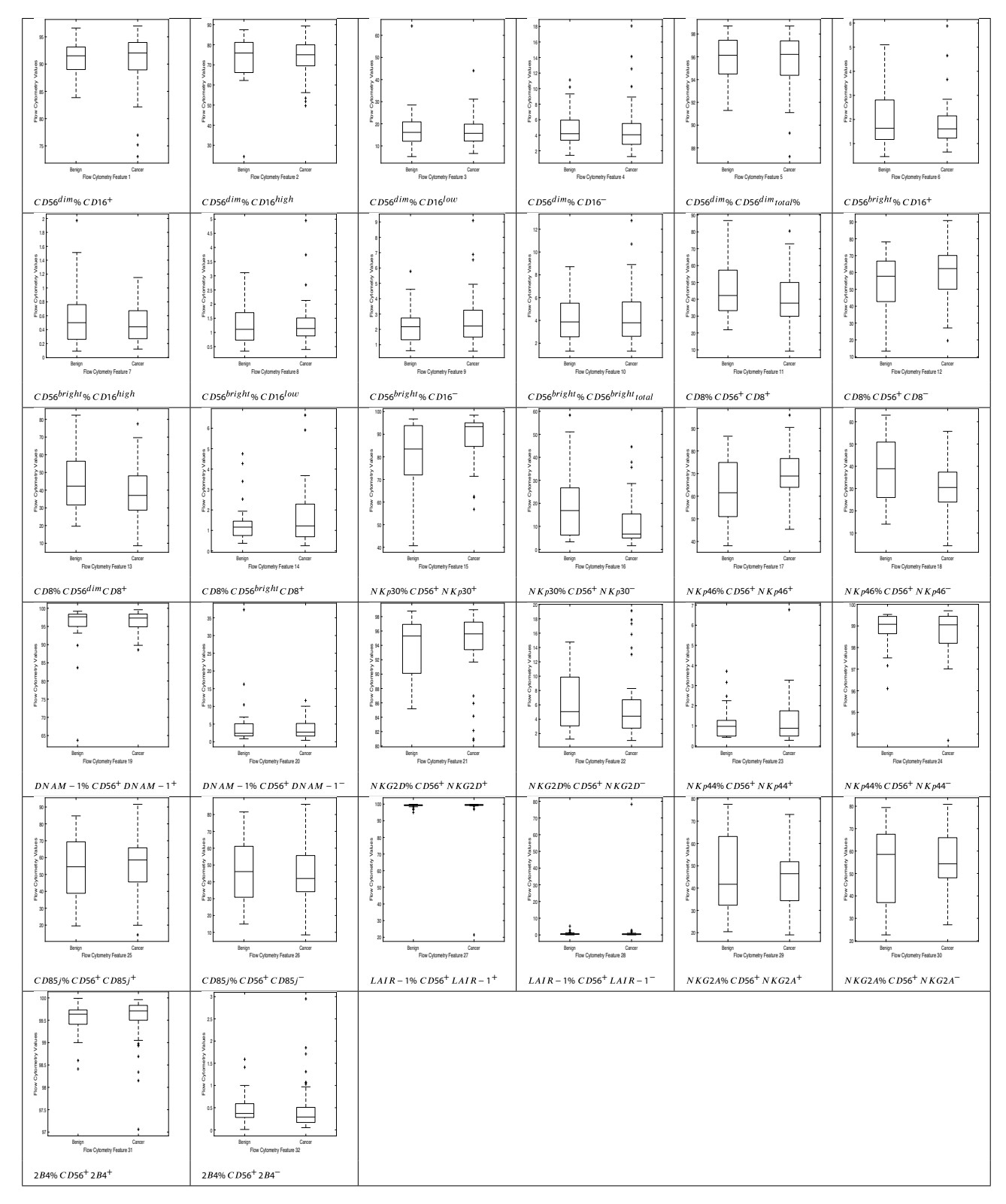

**Figure 1.** NK cell phenotypic features in men with benign prostate disease and patients with prostate cancer. Boxplots represent the flow cytometry values of each feature for patients with benign disease and with prostate cancer.

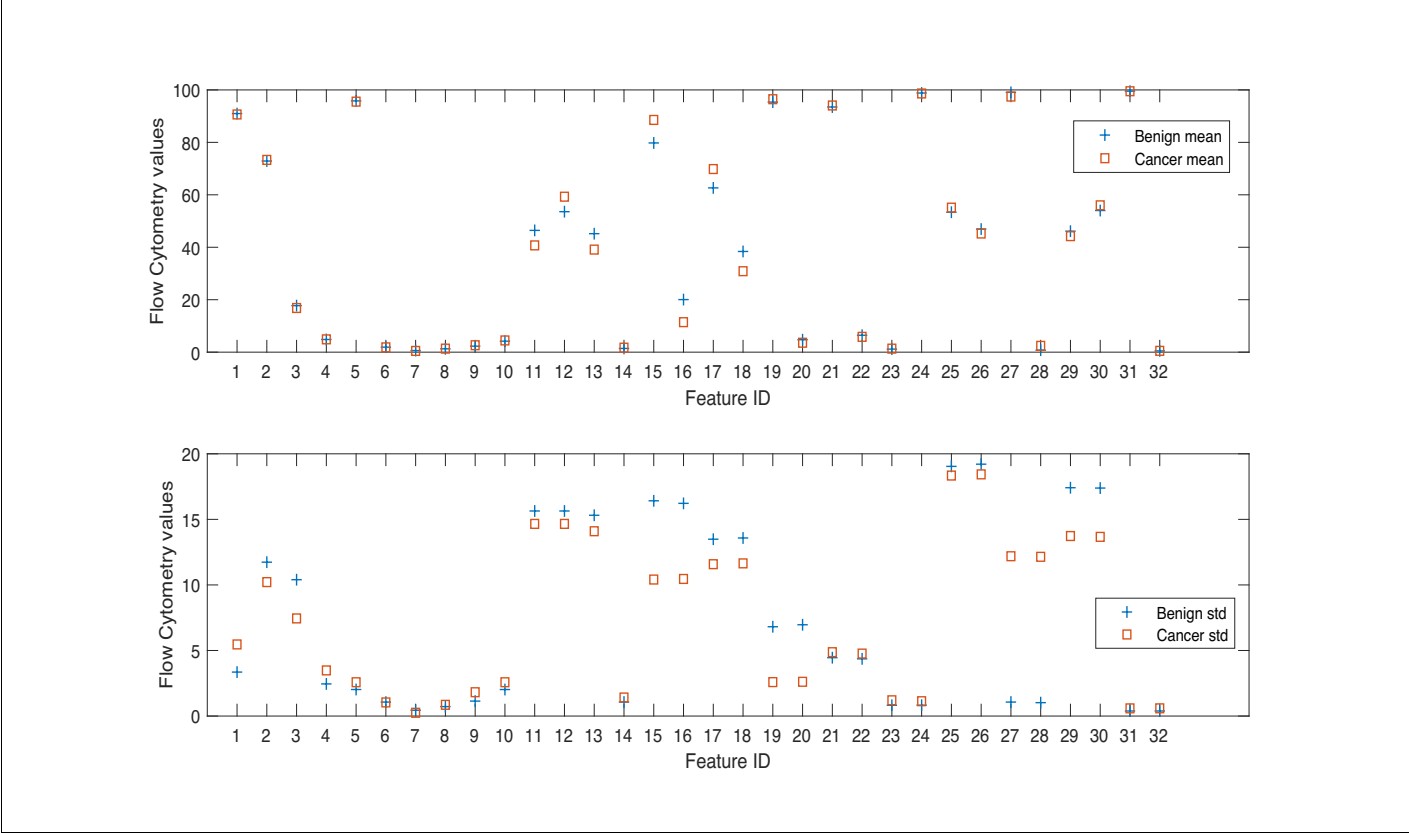

**Figure 2.** Mean and standard deviation values of flow cytometry features.

ID28: $CD56^+LAIR-1^-$. Referring back to *Figure 3* and the correlation values between the selected features 2, 20, 27, 28, 14, 15, 16, 17, it is shown that these features do not have a strong positive correlation. There is a strong negative correlation between features 27 and 28, but we decided to keep both features since these were selected by the feature selection method.

The next step in the analysis involves evaluating the predictive performance of the feature subsets returned by the statistical test and by the GA. The features identified from the statistical and GA approaches were input into the proposed Ensemble Subspace kNN classifier to determine whether it can learn these features and discriminate between the presence of benign prostate disease and prostate cancer. For transparency of the machine learning model, it was important to keep the predictor selection and machine learning processes separate. The feature selection algorithm identified a set of novel NK cell phenotypic features for diagnosing the presence of prostate cancer which will be used to construct a transparent prediction tool.

## Distinguishing between benign prostate disease and prostate cancer: machine learning

This section describes the outcome of experiments that were performed to determine the predictive performance of various feature subsets using the Ensemble Subspace kNN model, which was designed for the task. Machine learning classifiers that are constructed using small training sets have a large variance which means that the estimate of the target function will change if different training data are used (*Skurichina and Duin, 2002*). It is therefore expected, and normal, that classifiers will exhibit some variance. This means that small changes in input variable values can result in very different classification rules. To ensure that the proposed approach does not suffer from low variance, we evaluated the performance of the classifier using the 10-fold cross-validation approach which was repeated 30 times, for which the average and standard deviation of each run were recorded. Multiple runs of 10-fold cross-validation are performed using different partitions (i.e. folds), and the validation results are averaged over the runs to estimate a final predictive model. Each run of the cross-

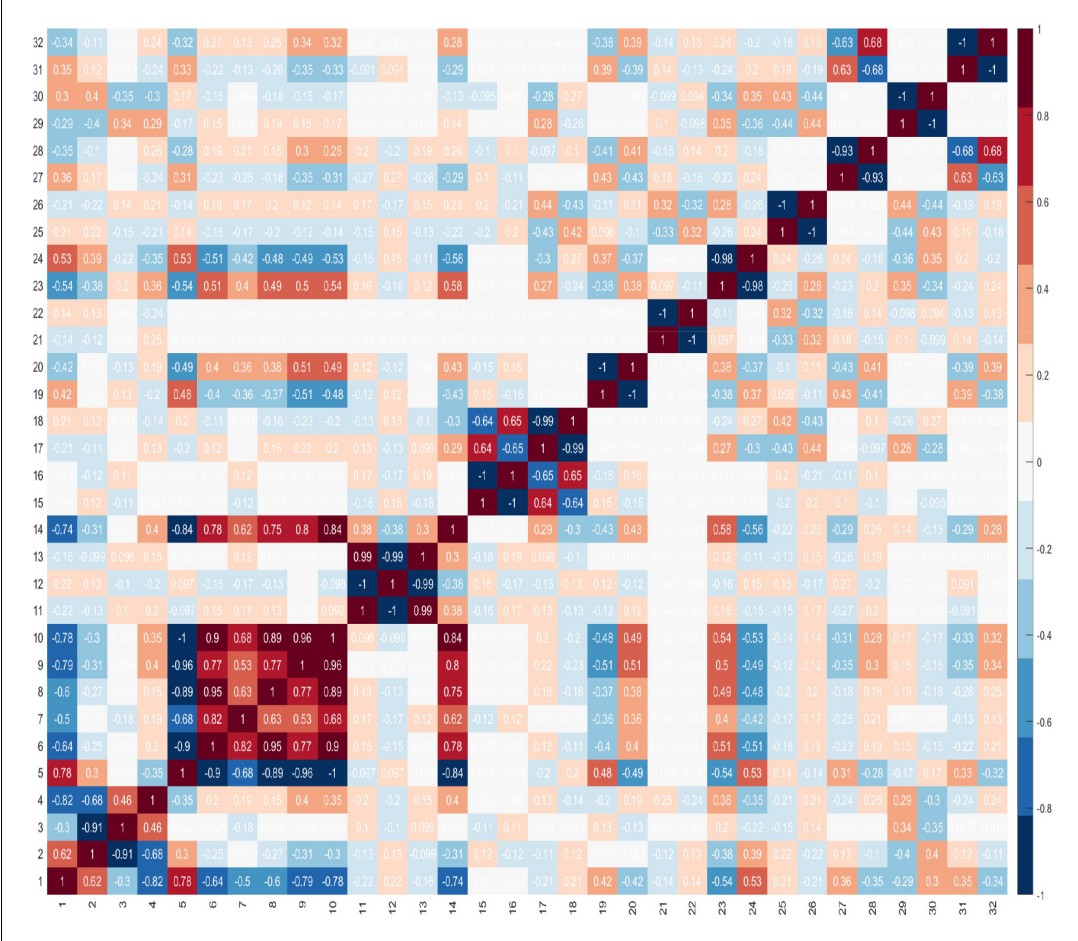

**Figure 3.** Correlations between features.

validation involves randomly partitioning a sample of data into complementary subsets, for which one subset is used as the training set, and the other is used as the validation subset. Cross validation randomly partitions the dataset into training and validation sets to limit overfitting problems, and to provide an insight into how the model will generalise to an independent dataset which was not previously seen by the model. A random seed generator was used to generate a different sequence of values each time the k-fold was run, and this was reseeded using a seed that was created using the current time. It is normal that a classifier returns a different validation accuracy in each fold and run, since it is training and validating on different samples. The aim is to create a low variance classifier, meaning that the results of each validation test are close together. The closer the results of each validation test, the more robust the classifier. To evaluate the predictive performance of various feature subsets derived from the computational and statistical feature selection approaches, each of these feature subsets was input into an Ensemble Subspace kNN classifier. Applying 10-fold validation resulted in 10 different partitions of the dataset of approximately 64 randomly selected samples for training and 7 randomly selected samples for validation in each partition (1 dataset comprising 63 training cases and 8 validation cases; and 9 datasets comprising 64 validation cases and 7 validation cases). All samples went through validation at some point during the evaluations. We consider 10-fold cross validation to be suitable given the small size of the dataset and the fact that sufficient samples are needed during the training process.

*Table 5* shows the results of the comparison when running the 10-fold validation 30 times using six sets of features: 1) the four features selected by the GA; 2) the four features which were returned by the Kruskal-Wallis statistical test (STAT); 3) combined features selected by the GA and the statistical test (GA+STAT); 4) PSA values combined with features selected by the GA and the statistical test

**Table 2.** Tests of normality results.

**Tests of normality**

| | NK cell values | | Kolmogorov-Smirnova | | | Shapiro-Wilk | | |
|---|---|---|---|---|---|---|---|---|
| | | | Statistic | df | Sig. | Statistic | df | Sig. |
| 1 | $CD56^{dim}$ | $CD16^+$ | 0.15 | 71.00 | 0.00 | 0.85 | 71.00 | 0.00 |
| 2 | $CD56^{dim}$ | $CD16^{high}$ | 0.11 | 71.00 | 0.03 | 0.89 | 71.00 | 0.00 |
| 3 | $CD56^{dim}$ | $CD16^{low}$ | 0.17 | 71.00 | 0.00 | 0.79 | 71.00 | 0.00 |
| 4 | $CD56^{dim}$ | $CD16^-$ | 0.19 | 71.00 | 0.00 | 0.82 | 71.00 | 0.00 |
| 5 | $CD56^{dim}$ | $CD56^{dim}total\%$ | 0.15 | 71.00 | 0.00 | 0.91 | 71.00 | 0.00 |
| 6 | $CD56^{bright}$ | $CD16^+$ | 0.13 | 71.00 | 0.00 | 0.88 | 71.00 | 0.00 |
| 7 | $CD56^{bright}$ | $CD16^{high}$ | 0.15 | 71.00 | 0.00 | 0.87 | 71.00 | 0.00 |
| 8 | $CD56^{bright}$ | $CD16^{low}$ | 0.14 | 71.00 | 0.00 | 0.85 | 71.00 | 0.00 |
| 9 | $CD56^{bright}$ | $CD16^-$ | 0.16 | 71.00 | 0.00 | 0.86 | 71.00 | 0.00 |
| 10 | $CD56^{bright}$ | $CD56^{bright}total$ | 0.15 | 71.00 | 0.00 | 0.91 | 71.00 | 0.00 |
| 11 | $CD8$ | $CD56^+CD8^+$ | 0.10 | 71.00 | 0.06 | 0.98 | 71.00 | 0.17 |
| 12 | $CD8$ | $CD56^+CD8^-$ | 0.10 | 71.00 | 0.06 | 0.98 | 71.00 | 0.17 |
| 13 | $CD8$ | $CD56^{dim}CD8^+$ | 0.09 | 71.00 | 0.20* | 0.98 | 71.00 | 0.24 |
| 14 | $CD8$ | $CD56^{bright}CD8^+$ | 0.19 | 71.00 | 0.00 | 0.82 | 71.00 | 0.00 |
| 15 | $NKp30$ | $CD56^+NKp30^+$ | 0.21 | 71.00 | 0.00 | 0.81 | 71.00 | 0.00 |
| 16 | $NKp30$ | $CD56^+NKp30^-$ | 0.21 | 71.00 | 0.00 | 0.81 | 71.00 | 0.00 |
| 17 | $NKp46$ | $CD56^+NKp46^+$ | 0.08 | 71.00 | 0.20* | 0.98 | 71.00 | 0.52 |
| 18 | $NKp46$ | $CD56^+NKp46^-$ | 0.07 | 71.00 | 0.20* | 0.99 | 71.00 | 0.57 |
| 19 | $DNAM-1$ | $CD56^+DNAM-1^+$ | 0.23 | 71.00 | 0.00 | 0.56 | 71.00 | 0.00 |
| 20 | $DNAM-1$ | $CD56^+DNAM-1^-$ | 0.23 | 71.00 | 0.00 | 0.55 | 71.00 | 0.00 |
| 21 | $NKG2D$ | $CD56^+NKG2D^+$ | 0.19 | 71.00 | 0.00 | 0.84 | 71.00 | 0.00 |
| 22 | $NKG2D$ | $CD56^+NKG2D^-$ | 0.18 | 71.00 | 0.00 | 0.85 | 71.00 | 0.00 |
| 23 | $NKp44$ | $CD56^+NKp44^+$ | 0.18 | 71.00 | 0.00 | 0.76 | 71.00 | 0.00 |
| 24 | $NKp44$ | $CD56^+NKp44^-$ | 0.17 | 71.00 | 0.00 | 0.78 | 71.00 | 0.00 |
| 25 | $CD85j$ | $CD56^+CD85j^+$ | 0.11 | 71.00 | 0.05 | 0.96 | 71.00 | 0.02 |
| 26 | $CD85j$ | $CD56^+CD85j^-$ | 0.10 | 71.00 | 0.07 | 0.96 | 71.00 | 0.02 |
| 27 | $LAIR-1$ | $CD56^+LAIR-1^+$ | 0.43 | 71.00 | 0.00 | 0.14 | 71.00 | 0.00 |
| 28 | $LAIR-1$ | $CD56^+LAIR-1^-$ | 0.43 | 71.00 | 0.00 | 0.14 | 71.00 | 0.00 |
| 29 | $NKG2A$ | $CD56^+NKG2A^+$ | 0.09 | 71.00 | 0.20* | 0.97 | 71.00 | 0.11 |
| 30 | $NKG2A$ | $CD56^+NKG2A^-$ | 0.08 | 71.00 | 0.20* | 0.97 | 71.00 | 0.10 |
| 31 | $2B4$ | $CD56^+2B4^+$ | 0.23 | 71.00 | 0.00 | 0.75 | 71.00 | 0.00 |
| 32 | $2B4$ | $CD56^+2B4^-$ | 0.23 | 71.00 | 0.00 | 0.75 | 71.00 | 0.00 |

*. This is a lower bound of the true significance.

Those values in bold are of those features whose data is normally distributed.

If the $p > 0.05$, we can accept the null hypothesis, that there is no statistically significant difference between the data and the normal distribution, hence we can presume that the data of those features are normally distributed.

If the $p < 0.05$, we can reject the null hypothesis because there is a statistically significant difference between the data and the normal distribution, hence we can presume that the data of those features are not normally distributed.

(PSA+GA+STAT); 5) PSA values alone as a predictor (PSA); and 6) using all 32 features (All features). The averages of the Area Under the Curve (AUC), Optimal ROC Point (ORP) False Positive Rate (FPR) of the AUC, ORP True Positive Rate (TPR) of the AUC, and Accuracy (ACC) of each fold are provided. The last column of *Table 5* shows the Rank of each model, where 1 is the best model and

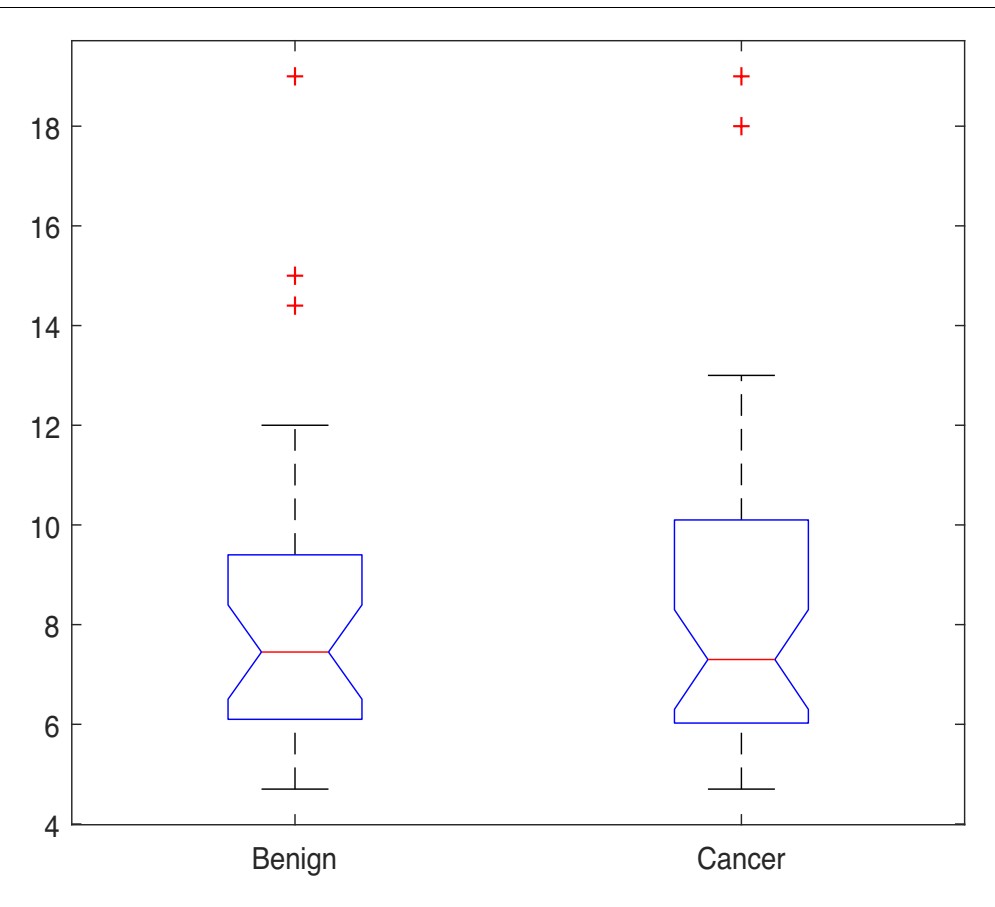

**Figure 4.** PSA values by group.

6 is the worst. The results of each k-fold were averaged, and these average values are plotted in the box plot shown in *Figure 5*. As shown in *Table 5*, combining the features selected by the GA ID2: $CD56^{dim}CD16^{high}$, ID20: $CD56^{+}DNAM-1^{-}$, ID27: $CD56^{+}LAIR-1^{+}$, ID28: $CD56^{+}LAIR-1^{-}$; with the four features which were returned by the Kruskal-Wallis statistical test as features with values which were statistically significant between individuals with benign prostate disease and patients with prostate cancer, ID14: $CD56^{bright}CD8^{+}$, ID15: $CD56^{+}NKp30^{+}$, ID16: $CD56^{+}NKp30^{-}$, ID17: $CD56^{+}NKp46^{+}$ yielded the highest classification accuracy, with AUC = 0.818, ORP FPR = 0.201, ORP TPR = 0.836 and Accuracy = 0.821. PSA values input into the classifier resulted in weak classification performance, AUC = 0.698, ORP FPR = 0.217, ORP TPR = 0.609, and Accuracy = 0.692. Although PSA is used as a screening test in clinical practice for identifying prostate cancer in men, it is the weakest of all the predictors. Importantly, predictive accuracy improved when PSA is combined with GA+STAT flow cytometry features (PSA+GA+STAT): AUC = 0.812, ORP FPR = 0.208, ORP TPR = 0.832, and ACC = 0.815. Combining PSA with the NK cell phenotypic fingerprint increased accuracy by +0.123 points when compared to using PSA alone.

The closer the standard deviation value is to 0 the less spread out are the results across the 30 runs, and hence the classifier variability is low (see *Table 5*). This results in a low variance classifier. A low standard deviation indicates that the data points tend to be close to the mean (also called the expected value) of the set, whereas a high standard deviation indicates that the data points are spread out over a wider range of values. Observing the data shown in *Table 5* and *Figure 5* for each evaluation measure (i.e. AUC, ORP TPR, ORP FPR, Accuracy (ACC)), the aim is to have a high AUC and low Std.; low ORP FPR and low Std.; high ORP TPR and low Std.; and high Accuracy and low Std. The results show that the classifier achieved the best performance when using the GA+STAT input and the results using k-fold across the 30 runs returned the lowest mean standard deviation

**Table 3.** Results of the Kruskal-Wallis test.

|  |  |  | Chi-Sq.($\chi^2$) | Asy. sig. p value |
|---|---|---|---|---|
|  | PSA |  | 0 | 0.949 |
|  | NK cells |  |  |  |
| 1 | $CD56^{dim}$ | $CD16^+$ | 0.001 | 0.981 |
| 2 | $CD56^{dim}$ | $CD16^{high}$ | 0.069 | 0.793 |
| 3 | $CD56^{dim}$ | $CD16^{low}$ | 0.555 | 0.456 |
| 4 | $CD56^{dim}$ | $CD16^-$ | 0.033 | 0.857 |
| 5 | $CD56^{dim}$ | $CD56^{dim}total\%$ | 0.063 | 0.802 |
| 6 | $CD56^{bright}$ | $CD16^+$ | 0.836 | 0.361 |
| 7 | $CD56^{bright}$ | $CD16^{high}$ | 0.201 | 0.654 |
| 8 | $CD56^{bright}$ | $CD16^{low}$ | 0.106 | 0.744 |
| 9 | $CD56^{bright}$ | $CD16^-$ | 0.030 | 0.861 |
| 10 | $CD56^{bright}$ | $CD56^{bright}total$ | 2.415 | 0.120 |
| 11 | $CD8$ | $CD56^+CD8^+$ | 2.415 | 0.120 |
| 12 | $CD8$ | $CD56^+CD8^-$ | 2.849 | 0.091 |
| 13 | $CD8$ | $CD56^{dim}CD8^+$ | 0.417 | 0.518 |
| 14 | $CD8$ | $CD56^{bright}CD8^+$ | 7.230 | **0.007** |
| 15 | $NKp30$ | $CD56^+NKp30^+$ | 7.106 | **0.008** |
| 16 | $NKp30$ | $CD56^+NKp30^-$ | 4.638 | **0.031** |
| 17 | $NKp46$ | $CD56^+NKp46^+$ | 5.179 | **0.023** |
| 18 | $NKp46$ | $CD56^+NKp46^-$ | 0.001 | 0.981 |
| 19 | $DNAM-1$ | $CD56^+DNAM-1^+$ | 0.001 | 0.972 |
| 20 | $DNAM-1$ | $CD56^+DNAM-1^-$ | 0.293 | 0.588 |
| 21 | $NKG2D$ | $CD56^+NKG2D^+$ | 0.325 | 0.568 |
| 22 | $NKG2D$ | $CD56^+NKG2D^-$ | 0.033 | 0.857 |
| 23 | $NKp44$ | $CD56^+NKp44^+$ | 0.072 | 0.789 |
| 24 | $NKp44$ | $CD56^+NKp44^-$ | 0.049 | 0.825 |
| 25 | $CD85j$ | $CD56^+CD85j^+$ | 0.072 | 0.789 |
| 26 | $CD85j$ | $CD56^+CD85j^-$ | 2.135 | 0.144 |
| 27 | $LAIR-1$ | $CD56^+LAIR-1^+$ | 1.343 | 0.247 |
| 28 | $LAIR-1$ | $CD56^+LAIR-1^-$ | 0.060 | 0.807 |
| 29 | $NKG2A$ | $CD56^+NKG2A^+$ | 0.072 | 0.789 |
| 30 | $NKG2A$ | $CD56^+NKG2A^-$ | 0.879 | 0.348 |
| 31 | $2B4$ | $CD56^+2B4^+$ | 0.890 | 0.346 |
| 32 | $2B4$ | $CD56^+2B4^-$ | 0.890 | 0.346 |

and hence the least variability in the results. The results reveal that using the GA+STAT predictors delivers a more reliable classification model with regards to training and validation on new data which will be generated in the future using the prediction model.

## Importance of findings

The GA+STAT prediction model achieved the best performance, in that the ORP FPR was the lowest, and the AUC, ORP TPR, and Accuracy (ACC) were the highest compared to the other prediction models. The experimental results are promising and the proposed prediction model is expected to achieve even higher classification accuracy in identifying the presence of prostate cancer in asymptomatic individuals with PSA levels < 20 ng ml$^{-1}$ based on peripheral blood NK cell phenotypic

**Table 4.** Results of the Genetic Algorithm when searching for the best subset of features.

| λ | No. different comb | Comb. with highest freq. | Freq. of comb. | Relative freq. (%) |
|---|---|---|---|---|
| 2 | 3 | 17,28 | 16 | 53.3 |
| 3 | 2 | 17,27,29 | 23 | 76.7 |
| 4 | 1 | 2,20,27,28 | 30 | 100.0 |
| 5 | 2 | 3,20,27,28,32 | 29 | 96.7 |
| 6 | 2 | 3,7,20,27,28,32 | 26 | 86.7 |
| 7 | 3 | 3,7,20,23,27,28,32 | 24 | 80.0 |
| 8 | 4 | 3,7,20,22,23,27,28,32 | 19 | 63.3 |
| 9 | 3 | 3,7,19,20,22,23,27,28,32 | 24 | 80.0 |
| 10 | 3 | 2,3,7,19,20,22,23,27,28,32 | 21 | 70.0 |

profiles as more data become available in the future. *Table 5* shows the performance of the classifier when using various feature subsets. When using the GA+STAT features, the AUC is higher, and FPR is lower (this is an important distinction) than when using all features or the other alternative feature subsets. The most important aspect is that better performance was achieved using a much smaller set of biomarkers (features), which indicates that we have identified a fingerprint for detecting the presence of prostate cancer in asymptomatic men with PSA levels < 20 ng ml$^{-1}$ which is indeed significant from a clinical perspective. Feature selection is important, as the fundamental aim of this project is to develop a subset of phenotypic biomarkers that is smaller than the original set of biomarkers (i.e. 32 biomarkers in total) which can confidently identify the presence of prostate cancer. Ultimately, the approach will be embedded into a software application to be used by clinicians, and the aim is to create an interface that requires the clinician to input a few values (features), that is 8 instead of 32. Importantly, identifying a small subset of 8 features which is needed for detecting the presence of prostate cancer, results in the construction of an explainable disease detection and categorization model. Working with a small set of the most promising biomarkers provides a better understanding of the disease and allows cancer immunobiologists and clinicians to focus on performing further laboratory evaluations using the specific subset of biomarkers, in a more cost effective and less time-consuming manner.

**Table 5.** Naming of the models includes the feature selection method (GA) combined with the proposed Ensemble Subspace kNN classifier.
Validation results are presented at k = 10 fold cross validation.

| | | Results of 10-fold cross validation over 30 runs | | | | | |
|---|---|---|---|---|---|---|---|
| | | AUC | ORP FPR | ORP TPR | ACC | Mean std. | Rank |
| GA | Mean | 0.776 | 0.296 | 0.833 | 0.781 | | 4 |
| | Std. | 0.024 | 0.065 | 0.026 | 0.023 | 0.035 | |
| STAT | Mean | 0.769 | 0.303 | 0.828 | 0.774 | | 5 |
| | Std. | 0.022 | 0.057 | 0.023 | 0.021 | 0.031 | |
| GA+STAT | Mean | 0.818 | 0.201 | 0.836 | 0.821 | | 1 |
| | Std. | 0.021 | 0.027 | 0.021 | 0.020 | 0.022 | |
| PSA+GA+STAT | Mean | 0.812 | 0.208 | 0.832 | 0.815 | | 2 |
| | Std. | 0.020 | 0.031 | 0.018 | 0.019 | 0.022 | |
| PSA | Mean | 0.698 | 0.217 | 0.609 | 0.692 | | 6 |
| | Std. | 0.022 | 0.025 | 0.043 | 0.020 | 0.028 | |
| All features | Mean | 0.812 | 0.213 | 0.836 | 0.815 | | 3 |
| | Std. | 0.022 | 0.035 | 0.021 | 0.021 | 0.025 | |

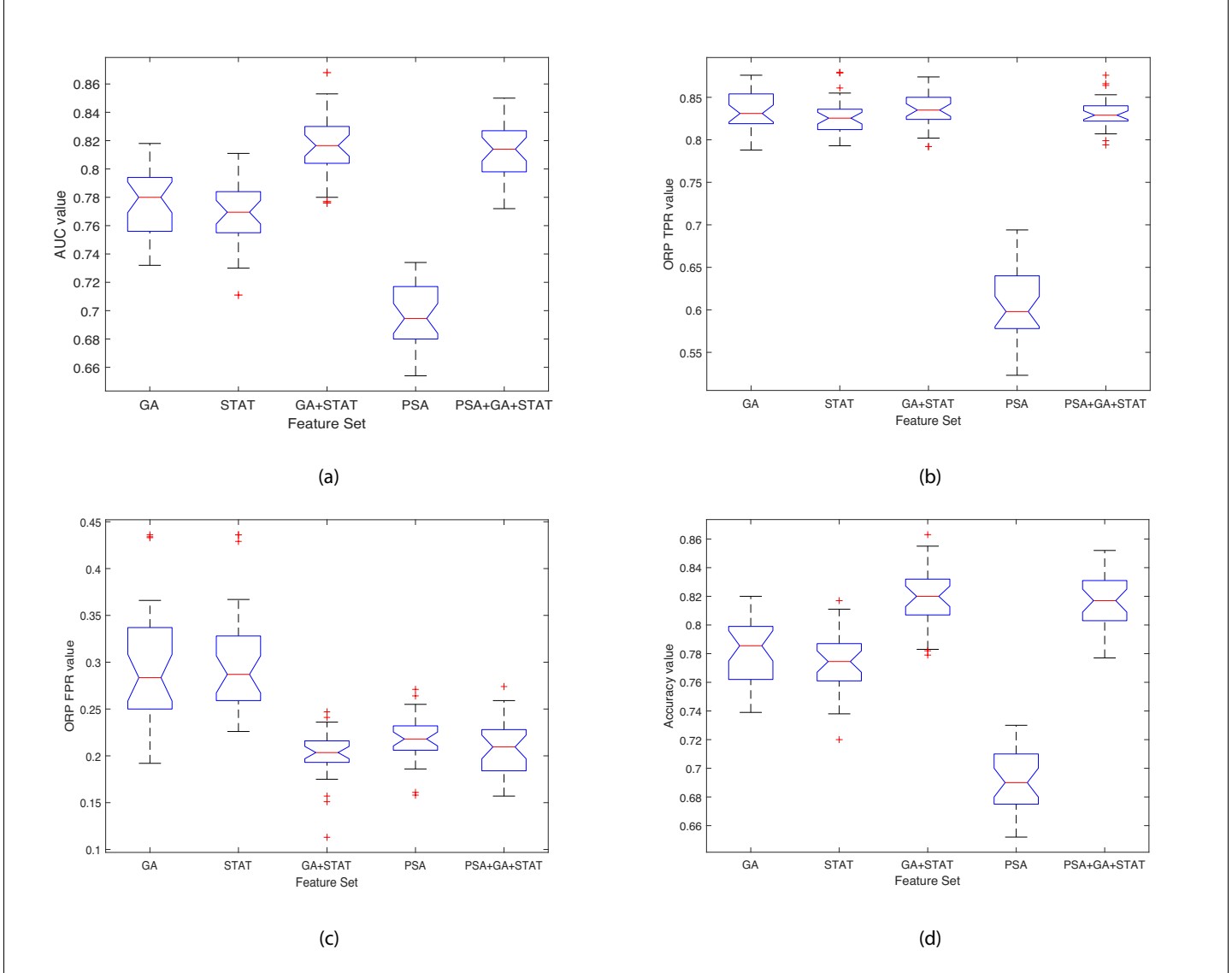

**Figure 5.** Boxplots illustrating the performance of the proposed model using various feature sets. (**a**) Average AUC values, (**b**) Average Optimal ROC points (TPRs), (**c**) Average Optimal ROC points (FPRs), (**d**) Average Accuracy values. Each box plot contains 30 points, where each point is the average performance evaluation value (i.e. AUC, ORP TPR, ORP FPR, Accuracy) from one 10-fold run using the various feature sets.

## Comparing the performance of the proposed ensemble subspace kNN classifier with alternative classifiers

The experiments discussed thus far utilised a machine learning model comprised of an Ensemble of kNN learners (see Section 'Proposed Ensemble Learning Classifier for the task of Predicting Prostate Cancer'). We then undertook experiments to determine the impact of using the proposed Ensemble method over conventional machine learning classifiers: simple kNN; Support Vector Machine; and Naive Bayes models. The last column of *Table 6* shows the difference in the performance of the methods. The proposed method, denoted as EkNN, returned better performance than all other alternative classifiers. EkNN also returned the lowest Standard Deviation values and these are an indicator of a more stable and reliable model since the average values are clustered closely around the mean. SVM-linear returned the highest ORP TPR; however, the higher ORP FPR, higher Std. values, the low AUC, and low Accuracy values suggest that this model is worse than the proposed EkNN. Naive Bayes was the least efficient classifier, and although it returned the lowest ORP FPR, it

**Table 6.** Comparing the performance of the proposed Ensemble Subspace kNN model against conventional machine learning models when using the GA+STAT feature set.
Results of 10-fold cross validation over 30 runs.

**Proposed ensemble subspace kNN (EkNN) model**
**(No. of learners (NL): 30; Subspace Dimension (SD): 16)**

| Parameters | | AUC | ORP FPR | ORP TPR | ACC | |
|---|---|---|---|---|---|---|
| NL: 30, SD:16 | Mean | 0.818 | 0.201 | 0.836 | 0.821 | |
| | Std. | 0.021 | 0.027 | 0.021 | 0.020 | |
| **Simple kNN model (Distance: Euclidean)** | | | | | | |
| k | | AUC | ORP FPR | ORP TPR | ACC | Acc. Diff. (EkNN vs. kNN) |
| 2 | Mean | 0.768 | 0.241 | 0.730 | 0.751 | +0.070 |
| | Std. | 0.119 | 0.160 | 0.393 | 0.128 | −0.108 |
| 5 | Mean | 0.778 | 0.300 | 0.833 | 0.783 | +0.038 |
| | Std. | 0.107 | 0.265 | 0.103 | 0.103 | −0.083 |
| 10 | Mean | 0.753 | 0.371 | 0.845 | 0.758 | +0.063 |
| | Std. | 0.137 | 0.350 | 0.120 | 0.131 | −0.111 |
| **Support Vector Machine models** | | | | | | |
| Kernel | | AUC | ORP FPR | ORP TPR | ACC | Acc. Diff. (EkNN vs. SVM) |
| Linear | Mean | 0.782 | 0.342 | 0.860 | 0.784 | +0.037 |
| | Std. | 0.126 | 0.352 | 0.110 | 0.120 | −0.100 |
| Gaussian | Mean. | 0.808 | 0.353 | 0.876 | 0.799 | +0.022 |
| | Std. | 0.112 | 0.416 | 0.107 | 0.111 | −0.091 |
| **Naive Bayes model** | | | | | | |
| Predictor distributions | | AUC | ORP FPR | ORP TPR | ACC | Acc. Diff. (EkNN vs. Naïve Bayes) |
| Normal | Mean. | 0.695 | 0.132 | 0.455 | 0.662 | +0.159 |
| | Std. | 0.169 | 0.163 | 0.493 | 0.181 | −0.161 |

also returned the lowest ORP TPR, lowest AUC and Accuracy values; and its Std. values were also higher than those of the EkNN model.

## Statistically significant differences in predictive performance when using various feature subsets

The next step in the analysis is to determine whether statistically significant differences exist between the average AUC performance values of the classifier when using the various feature subsets, for which Friedman's two-way Analysis of Variance (ANOVA) test was used. It was also important to observe whether including the PSA test values significantly strengthens the diagnostic accuracy and capacity. The average k-fold values across the 30 runs for each feature set were computed. A matrix C was derived which holds the results of the classifier when using one of five feature subsets. Friedman's chi-square statistic compares the mean values of the columns of matrix C. The test returned a statistically significant difference in the AUC predictive performance depending on which type of feature subset was input into the classifier, $\chi^2(4) = 106.55$, $p = 3.968E − 22$. This suggests that the mean AUC ranks of at least one feature subset are significantly different than the others. The mean ranks were as follows: GA = 12.050, STAT = 10.733, GA+STAT = 20.283, PSA = 3.067, PSA+GA+STAT = 18.867. A post hoc test was run alongside the Friedman test to pinpoint which feature subsets differ from each other. Post hoc analysis using a Bonferroni correction was used to reduce the likelihood of erroneously declaring a statistically significant due to multiple comparisons (a Type I error). *Table 7* shows the results of multiple comparisons and adjusted p

**Table 7.** Ad hoc test results.

| | Ad hoc test | | | | | |
|---|---|---|---|---|---|---|
| | Group 1 | Group 2 | LI 95% | Diff. betw.means | UI 95% | p |
| 1 | GA | STAT | −12.658 | 1.317 | 15.292 | 1.000 |
| 2 | GA | GA+STAT | −22.208 | −8.233 | 5.742 | 0.525 |
| 3 | GA | PSA | −4.992 | 8.983 | 22.958 | 0.344 |
| 4 | GA | PSA+GA+STAT | −20.792 | −6.817 | 7.158 | 1.000 |
| 5 | STAT | GA+STAT | −23.525 | −9.550 | 4.425 | 0.245 |
| 6 | STAT | PSA | −6.308 | 7.667 | 21.642 | 0.710 |
| 7 | STAT | PSA+GA+STAT | −22.108 | −8.133 | 5.842 | 0.555 |
| 8 | GA+STAT | PSA | 3.242 | 17.217 | 31.192 | 0.001 |
| 9 | GA+STAT | PSA+GA+STAT | −12.558 | 1.417 | 15.392 | 1.000 |
| 10 | PSA | PSA+GA+STAT | −29.775 | −15.800 | −1.825 | 0.002 |

The first two columns show the groups that are compared. The third and fifth columns show the lower and upper limits for 95% confidence intervals for the true mean difference. The fourth column shows the difference between the estimated group means. The sixth column contains the p-value for testing a hypothesis that the corresponding mean difference is equal to zero.

values. There were statistically significant differences between group 8 (GA+STAT vs. GA) and 10 (PSA vs. PSA+GA+STAT) (p=0.001). We can conclude that GA+STAT returned a significantly higher AUC than PSA, and the difference between their mean ranks is diff = 17.217. PSA returned a significantly lower AUC than PSA+GA+STAT (p=0.002), and the difference between their mean ranks is diff=-15.800.

## Comparing the best prediction models over 30 runs

With regard to constructing a model which has the potential to be used in clinical practice, it is necessary to finalise an initial prediction model, since the last experiment returned 30 different variations of each prediction model when using different training and validation data partitions. Those experiments were crucial in determining whether the prediction models (five models, a different one for each feature subset) suffer from low variance. We then observed the classification performance of each model for each run, to identify the highest performance achieved using a single 10-fold cross validation in any of the runs. This provides a way of comparing the performance of each prediction model as it would be used in the clinical setting. *Table 8* provides the results of the highest performing model, and the performance of the models is ranked (with 1 being the best model and 5 the worst model).

## Predicting low-/intermediate risk cancer vs. high-risk cancer

The continuing, significant clinical challenge resides in distinguishing men with low- or intermediate-risk prostate cancer which is unlikely to progress (for both of which 'active surveillance' is the most appropriate approach), from men with intermediate disease which is likely to progress and men with

**Table 8.** Results of the best prediction models created during the 30 runs.
Validation results are presented at k = 10 fold cross validation.

| | Best prediction model results | | | | |
|---|---|---|---|---|---|
| | AUC | ORP FPR | ORP TPR | Accuracy | Rank |
| GA | 0.818 | 0.192 | 0.829 | 0.820 | 3 |
| GA+STAT | 0.853 | 0.157 | 0.862 | 0.855 | 1 |
| PSA | 0.734 | 0.218 | 0.685 | 0.730 | 5 |
| PSA+GA+STAT | 0.844 | 0.175 | 0.864 | 0.848 | 2 |
| STAT | 0.811 | 0.227 | 0.85 | 0.817 | 4 |

high-risk prostate cancer (both of which require treatment). The diagnosis of men with low-risk or small volume intermediate-risk prostate cancer as having prostate cancer is unhelpful as these men will very rarely require treatment. The inappropriate assignment of men to potentially life-threatening invasive procedures and life-long surveillance for prostate cancer has significant psychological, quality of life, financial, and societal consequences. Furthermore, the definitive diagnosis of prostate cancer currently requires painful invasive biopsies with which is associated a risk of potentially life-threatening urosepsis in 5% of individuals. We, therefore, undertook experiments to train the proposed Ensemble Subspace kNN model to predict the D'Amico Risk Classification for those patients with prostate cancer (see subsection 'The cancer patients dataset used for building the risk prediction modelin Methods), in terms of Low/Intermediate (L/I) risk and High (H) risk disease using NK cell phenotypic data alone.

The Ensemble model was modified to take as input all 32 features (described in *Table 1*), and was trained to classify the disease in patients with prostate cancer as being L/I or H risk disease (see Figure 9 in Materials and methods). Hence, given a new patient record, which comprises of 32 inputs, the model predicts whether the patient is D'Amico L/I risk (not clinically significant) or H (clinically significant) risk. The flow charts in *Figure 6* illustrate the process to detect the presence and risk of prostate cancer and patient outcomes. Of those 54 patient records, a total of 10 randomly selected records (5 from the L/I group and 5 from the H group) were extracted from the dataset such that they can be used at the testing (mini clinical trial) stage. To ensure thorough experiments, a rigorous methodology was adopted. More specifically, a 10-fold cross validation method was adopted, and the experiments were run in 30 iterations, for which each iteration provided an average validation result across 10 folds. Each iteration consists of 10 different 'train and validation' data arrangements (hence 300 tests were carried out using a different mix of train and validation records). The 10 test records were input into each trained model (i.e. iteration) to predict their accuracy, and

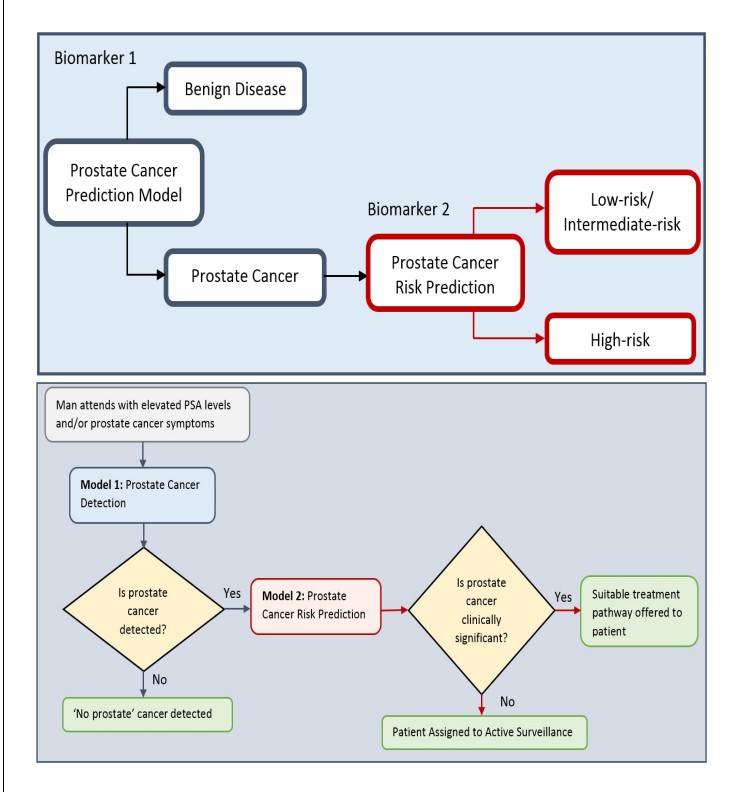

**Figure 6.** Flow charts illustrating the process to detect the presence and risk of prostate cancer and patient outcomes. Model 1: Distinguishes between men with benign prostate disease and prostate cancer; Model 2: predicts risk (in terms of clinical significance) in men identified as having prostate cancer in Stage 1. Note that Model 1 can detect prostate cancer in men with PSA < 20 ng ml$^{-1}$.

to evaluate the model when it is trained and validated using different variations of patient data. The model can highly accurately differentiate between L/I risk group and H risk group patients. The k-fold validation results across 30 iterations were AUC: 0.98(±0.03); FPR: 0.03(±0.05), TPR: 0.99 (±0.01), Accuracy: 0.99(±0.02); and results using the test set were AUC: 0.98(±0.03); FPR: 0.03 (±0.05), TPR: 0.99(±0.01), Accuracy: 0.97(±0.02). Accuracy has been near perfect in all iterations (i.e. using different train and validation data cases in each iteration). *Figure 7* illustrates the performance of the model obtained across the 30 runs during the k-fold cross validation and independent testing using the 10 patient samples. The results demonstrate that the proposed model predicts with near-perfect accuracy, the result of the D'Amico Risk Classification (L/I vs High) using NK cell phenotypic data alone, and without requiring the PSA, Gleason, and tumor stage data.

The dataset that was utilized to identify the biomarker (that comprised eight features) for detecting the presence of prostate cancer (i.e. benign prostate disease vs prostate cancer) in 71 men, and thus it was large enough to perform the combinatorial feature selection task for finding the best subset of features. The GA that was used for the combinatorial feature selection task is described in Section Computational Methods. Given that detecting the presence of prostate cancer and its risk if present are two different tasks, it is expected that the biomarkers for those tasks will be different since a different target is given to the GA (i.e. the target for the prostate cancer detection model comprises 0 (benign prostate disease) and 1 (prostate cancer) values; the target for the prostate cancer risk prediction model comprises 0 (L/I risk) and 1 (High risk) values). For the L/I vs H risk task, the dataset was small (n = 54 men (L/I = 36, H = 16)), and we could not perform the combinatorial feature selection task with confidence. Hence, it was decided to use the entire feature set for the risk prediction task. The results obtained from the risk prediction model were very promising as shown experimentally, and this provided the confidence to report these preliminary results. The combinatorial feature selection task to identify the best subset of features for the risk prediction task will be performed once a larger dataset is available.

Herein, we demonstrate that all 32 phenotypic features are required to distinguish between low/intermediate risk cancer (L/I) and high risk (H) cancer. However, we expect to be able to identify smaller subset(s) of these features as the datasets increase and the prediction model is retrained on the larger dataset. As indicated above, the generation and delivery of additional datasets is beyond the scope of this paper.

## Discussion

The clinical challenge in prostate cancer diagnosis resides in distinguishing men with low- or small volume intermediate-risk prostate cancer which is unlikely to progress (both require 'active surveillance') from men with intermediate disease which is likely to progress or high-risk disease (both of

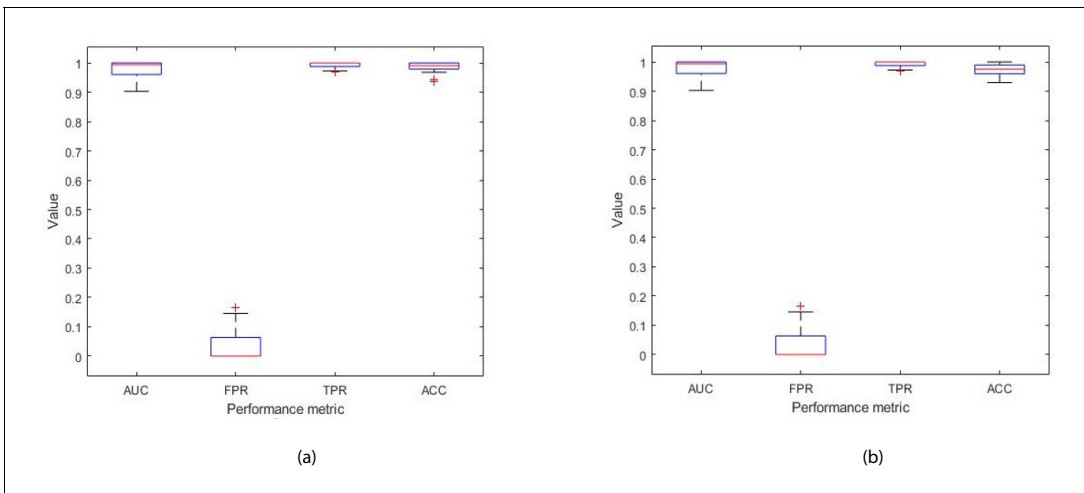

(a)

(b)

**Figure 7.** Each box plot contains 30 points, where each point is the average performance evaluation value (i.e. AUC, FPR, TPR, Accuracy (ACC)) from one 10-fold run during (a) k-fold validation results, and (b) independent testing results (i.e. using 10 patient records).

which require treatment). It is essential that men with low-risk prostate abnormalities are not diagnosed as having prostate cancer, as those with low-risk/grade disease do not require active treatment. Furthermore, unnecessarily labeling men as having prostate cancer can assign these men to life-long surveillance and have significant psychological, quality of life, financial and societal consequences. Recent findings from a decade-long study involving 415,000 British men (The Cluster Randomized Trial of PSA Testing for Prostate Cancer (CAP) Randomized Clinical Trial) have not supported single PSA testing for population-based screening and suggest that asymptomatic men should not be routinely tested to avoid unnecessary anxiety and treatment. It is therefore essential that new approaches for enabling more definitive, early detection of prostate cancer are developed. The reliable diagnosis of prostate cancer based on PSA levels alone is not possible and confirmation using invasive biopsies or other approaches such as MRI and biopsy are currently required. Although interest in the potential diagnostic capabilities of MRI scanning is developing, MRI cannot currently be used as a sole diagnostic as a positive MRI can be incorrect in approximately 25% of cases and a negative MRI can be incorrect in approximately 20% of cases *Ahmed et al., 2017*. Although the findings from the CAP study do not support using the PSA test as an approach for population-based screening, combining PSA measurements with other approaches that either identify individuals for additional testing or strengthen the capacity to diagnose prostate cancer have significant merit, and it is based on this concept that the current study has been performed. The studies presented herein have focused on asymptomatic men with a PSA < 20 ng/ml, as men with a PSA level > 20 ng/ml are more likely to harbour prostate cancer and are thereby less likely to pose a clinical diagnostic quandary. In contrast, men with a PSA < 20 ng/ml pose a major problem because although only 30–40% of these men will have prostate cancer, all currently undergo potentially unnecessary invasive prostate biopsies to determine who has the disease. It is, therefore, this group of men for which the development of new and more accurate approaches for the early detection of cancer is a clear unmet clinical need, and for whom the benefits of such an approach will be most relevant and significant.

## Comparing results to the previous study

We have previously shown that incorporating peripheral blood immune phenotyping-based features into a computation-based prediction tool enables the better detection of prostate cancer and, furthermore, strengthens the accuracy of the PSA test in asymptomatic individuals having PSA levels < 20 ng/ml (*Cosma et al., 2017*). The phenotypic feature set which was shown to be discriminatory between benign disease and prostate cancer comprised $CD8^+CD45RA^-CD27^-CD28^-$ ($CD8^+$ Effector Memory cells), $CD4^+CD45RA^-CD27^-CD28^-$ ($CD4^+$ Effector Memory Cells), $CD4^+CD45RA^+CD27^-CD28^-$($CD4^+$ Terminally Differentiated Effector Memory Cells re-expressing CD45RA), $CD3^-CD19^+$ (B cells), $CD3^+CD56^+CD8^+CD4^+$ (NKT cells).

Using samples from the same cohort of asymptomatic individuals, herein we have further investigated the phenotype and function of NK cell subsets. Using a combination of statistical and computational feature selection approaches, we have identified a subset of eight phenotypic features $CD56^{dim}CD16^{high}$, $CD56^+DNAM-1^-$, $CD56^+LAIR-1^+$, $CD56^+LAIR-1^-$, $CD56^{bright}CD8^+$, $CD56^+NKp30^+$, $CD56^+NKp30^-$, $CD56^+NKp46^+$ which distinguish between the presence of benign prostate disease and prostate cancer. These features were used to implement a prediction model. The kNN machine learning approach developed in our previous study (*Cosma et al., 2017*) has been extended to an Ensemble of kNN learners to improve performance in identifying patterns in even more complex data. As was observed in our previous study, flow cytometry predictors significantly outperform the PSA test. The findings presented herein significantly reinforce our previous finding (*Cosma et al., 2017*) that complementing the PSA prediction model with a subset of flow cytometry-based phenotypic predictors can significantly increase the accuracy of the initial prostate cancer test and reduce misclassification. The performance of the prediction model which was built using the phenotypic 'signature' presented in our previous study $-CD8^+CD45RA^-CD27^-CD28^-$, $CD4^+CD45RA^-CD27^-CD28^-$, $CD4^+CD45RA^+CD27^-CD28^-$, $CD3^-CD19^+$, $CD3^+CD56^+CD8^+CD4^+$ (*Cosma et al., 2017*), is similar to the model which was built using the NK cell-based phenotypic signature presented herein, $CD56^{dim}CD16^{high}$, $CD56^+DNAM-1^-$, $CD56^+LAIR-1^+$, $CD56^+LAIR-1^-$, $CD56^{bright}CD8^+$, $CD56^+NKp30^+$, $CD56^+NKp30^-$, $CD56^+NKp46^+$. Specifically, the prediction model using the five flow cytometry features identified in *Cosma et al.,*

*2017* achieved Accuracy: 83.33% , AUC: 83.40%, ORP TPR: 82.93%, FPR: 16.13%, whereas the prediction model presented herein achieved AUC: 85.3%, ORP FPR: 15.7%, ORP TPR: 86.2%, Accuracy: 85.5%. Across the 30 runs the average performance of the prediction model presented herein is AUC: 81.8%, ORP TPR: 83.6%, FPR: 20.1%, Accuracy: 82.1%.

The difference in the performance of the model presented in the first study (*Cosma et al., 2017*) and the study described herein is a consequence of different data and prediction models being used in each study. Given that the phenotypic features that were used to create the prediction models were different, the studies resulted in different prediction models. In particular, the model presented previously (*Cosma et al., 2017*) was based on a kNN classifier, and herein the kNN classifier was extended to construct an Ensemble Subspace kNN method which comprised several kNN classifiers (see Figure 9). The dataset used herein was more complex, and it was therefore necessary to create a more complex classifier. At this point in the studies, it is not possible to determine which set of phenotypic features is better at identifying prostate cancer. However, it is evident that both approaches have significant promise. Since the publication of our previous study (*Cosma et al., 2017*), the model developed for that study was used to predict the outcomes of a further 20 new patients which were previously unseen by the prediction model. The model correctly identified the presence of prostate cancer in 19 of the 20 patients (data not shown).

Encouragingly, the prediction models generated in the study reported upon herein selected phenotypic features that are associated with the expression of activating receptors NKp30, NKp46, and DNAM-1 by NK cells. *Pasero et al., 2015* demonstrated that these activating receptors, in addition to NKG2D, are involved in the recognition of prostate cancer cell lines. Furthermore, they identified that the intensity of NKp30 and NKp46 expression on the surface of NK cells isolated from the peripheral blood of patients with metastatic prostate cancer was predictive of time to hormone (castration) resistance and overall survival. This suggests that our computational analysis is selecting phenotypic features that are of biological/clinical relevance. Thus far, our identification of disease predictive phenotypic immune features has been limited to effector immune populations (T, B, and NK cells). The responsiveness of these cells is known to be influenced by the presence of innate immune cell populations that can be polarized by the tumor toward an immunosuppressive state (*Vitale et al., 2014*; *Anderson et al., 2017*). Therefore, future studies will investigate the identification and inclusion of phenotypic features from innate immune subpopulations such as monocytes and neutrophils into prediction models to assess whether their inclusion enhances predictive capability and enables a better assessment of patient prognosis in line with the D'Amico Risk Classification.

The proposed machine learning model was adapted to predict the D'Amico Risk Classification of patients with prostate cancer using NK cell phenotypic data alone. Experiments with data from 54 patients revealed the significant potential of using the proposed machine learning model for determining if men with prostate cancer are in the low-/intermediate- or high-risk groups, without the need for additional clinical data (i.e. PSA, Gleason, clinical stage data). One limitation of the current study is that the small patient numbers required for low- and intermediate-risk patients to be grouped. Future work, for which additional sample collections are required, will train the model to separately predict low-, intermediate- and high-risk cancer. Future work involves collecting more patient samples to conduct further testing of the proposed machine learning models. In terms of future work from a computational perspective, once we have a larger patient dataset we plan to design deep learning models and compare their performance to the conventional machine learning model which was proposed in this paper.

## Potential impact

Currently available screening methods and tests for prostate cancer lack accuracy and reliability, the consequence of which is that many men unnecessarily undergo invasive tests such as biopsy and/or are misdiagnosed as having the disease. Furthermore, a biopsy involves removing samples of tissue from the prostate and it is an extremely uncomfortable procedure which also puts men at risk of developing life-threatening infections. As biopsy results are not definitive, there is a significant potential for misdiagnosis and over- and under- treatment. It is therefore essential that new non-invasive approaches such as blood tests that are more accurate than the Prostate Specific Antigen (PSA) test are developed to reduce misdiagnosis and unnecessary procedures. Misdiagnosis unnecessarily subjects many men to lifelong monitoring for prostate cancer which can have undesirable psychological and quality of life side-effects, as well as place a significant financial burden on the

NHS and other healthcare systems. This paper proposes a computerised model, which detects the presence of prostate cancer in men by analyzing immune system cells in the blood. The model uses the data from the blood tests and artificial intelligence-based computing (machine learning) to more accurately detect the presence of prostate cancer. A preliminary model has also been presented to detect the clinical risk that any prostate cancer which is present poses. The tool has two elements, the first detects whether a man has prostate cancer. If prostate cancer is detected, the second element will detect the clinical risk of the disease (low, intermediate, high) and thereby enable the clinician to decide whether the patient requires no further investigation/treatment ('watch and wait') or whether further investigation and treatment are required.

To our knowledge, these are the first studies to employ computational modeling of peripheral blood NK cell phenotyping data for the early detection of cancer and its clinical significance. They also illustrate the potential for this approach to decipher clinically relevant immune features that can distinguish between benign prostate disease and prostate cancer in asymptomatic individuals for whom the management and treatment strategy is unclear. Of translational importance is that our prediction models are interpretable, can be explained to patients and clinicians and can be continually refined and improved as data are collected.

The novelty of this approach is that it interrogates the immunological response to the tumour, not the tumour itself and that it requires a simple blood test (liquid biopsy). Based on current practice, we expect that this approach could avoid up to 70% of prostate biopsies, thereby sparing men with benign prostate disease or low-risk prostate cancer from unnecessary invasive procedures with which are associated significant side-effects. Furthermore, more accurate diagnosis would reduce the demands of healthcare provision and resources associated with treatment and continual surveillance, thereby reducing costs and improving healthcare. We envisage that, in the future, men with a mildly elevated PSA will also undergo an immune status test and those with a suspicion for significant prostate cancer will then undergo an MRI. Although the current study focuses on prostate cancer, its fundamental principles and approaches are highly likely to be applicable across many, if not all, cancer entities.

# Materials and methods

## Key resources table

| Reagent type (species) or resource | Designation | Source or reference | Identifiers | Additional information |
|---|---|---|---|---|
| Biological Sample | Hyclone fetal bovine serum (FBS) | GE Healthcare Life Sciences | SV30180.03 | |
| Antibody | Monoclonal mouse IgG1 kappa anti human DNAM-1 (CD226) (clone 11A8); FITC | BioLegend | 338304 | 5 µl per tube / $10^6$ cells |
| Antibody | Monoclonal mouse IgG1 kappa anti human NKG2D (CD314) (clone 1D11); PE | eBioscience | 12-5878-42 | 5 µl per tube / $10^6$ cells |
| Antibody | Monoclonal mouse IgG1 kappa anti human CD56 (clone N901); ECD (PE-Texas Red) | Beckman Coulter | A82943 | 2.5 µl per tube / $10^6$ cells |
| Antibody | Monoclonal mouse IgG1 kappa anti human CD16 (clone 3G8); PerCP-Cy5.5 | BioLegend | 302028 | 5 µl per tube / $10^6$ cells |
| Antibody | Monoclonal mouse IgG1 kappa anti human NKp46 (CD335) (clone 9E2); PE-Cy7 | BioLegend | 331916 | 5 µl per tube / $10^6$ cells |

*Continued on next page*

*Continued*

| Reagent type (species) or resource | Designation | Source or reference | Identifiers | Additional information |
|---|---|---|---|---|
| Antibody | Monoclonal mouse IgG1 kappa anti human NKp30 (CD337) (clone P30-15); Alexa Fluor 647 | BioLegend | 325212 | 5 µl per tube / $10^6$cells |
| Antibody | Monoclonal mouse IgG1 kappa anti human CD3 (clone UCHT1); Alexa Fluor 700 | BioLegend | 300424 | 2 µl per tube / $10^6$cells |
| Antibody | Monoclonal mouse IgG1 kappa anti human CD19 (clone HIB19); Alexa Fluor 700 | BioLegend | 302226 | 1 µl per tube / $10^6$cells |
| Antibody | Monoclonal mouse IgG1 kappa anti human CD8 (clone SK1); APC-Cy7 | BioLegend | 344714 | 2.5 µl per tube / $10^6$cells |
| Antibody | Monoclonal mouse IgG2b anti human CD85j (ILT2) (clone GHI/75); FITC | Miltenyi Biotec | 130-098-437 | 10 µl per tube / $10^6$cells |
| Antibody | Monoclonal mouse IgG1 kappa anti human LAIR-1 (CD305) (clone DX26); PE | BD Biosciences | 550811 | 20 µl per tube / $10^6$cells |
| Antibody | Monoclonal mouse IgG2b anti human NKG2A (CD159a) (clone Z199); PE-Cy7(PC7) | Beckman Coulter | B10246 | 20 µl per tube / $10^6$cells |
| Antibody | Monoclonal mouse IgG1 kappa anti human NKp44 (CD336) (clone P44-8); Alexa Fluor 647 | BioLegend | 325112 | 5 µl per tube / $10^6$cells |
| Antibody | Monoclonal mouse IgG1 kappa anti human 2B4 (CD244.2) (clone C1.7); FITC | BioLegend | 329506 | 5 µl per tube / $10^6$cells |
| Chemical Compound | LIVE/DEAD Fixable Violet Dead Stain | Thermo Fisher Scientific | L34955 | 1 µl in 1 µl |
| Chemical Compound | Novagen Benzonase Nuclease | Merck Millipore | 70664 | |
| Chemical Compound | CTL Wash Solution | Cellular Technology Limited | CTLW-010 | |
| Chemical Compound | Trypan Blue viability stain | Santa Cruz | sc-216028 | |
| Chemical Compound | Dimethyl sulfoxide (DMSO) | Santa Cruz | sc-202581 | |
| Chemical Compound | Calbiochem bovine serum albumin (BSA) | Merck Millipore | 2905-OP | |
| Chemical Compound | Sigma-Aldrich sodium azide | Merck Millipore | S8032 | |
| Chemical Compound | Sigma-Aldrich lithium heparin | Merck Millipore | H0878 | |
| Chemical Compound | Ficoll-Paque | GE Healthcare Life Sciences | 17-1440-03 | |
| Chemical Compound | Isoton II isotonic buffered saline solution | Beckman Coulter | 844 80 11 | |

*Continued on next page*

*Continued*

| Reagent type (species) or resource | Designation | Source or reference | Identifiers | Additional information |
|---|---|---|---|---|
| Chemical Compound | RPMI medium | Lonza | 12-167Q | |
| Chemical Compound | Phosphate Buffered Saline (PBS) | Lonza | 17-517Q | |
| Other | Leucosep tubes | Greiner Bio-One International | 227290 | |
| Software | Kaluza v1.3 | Beckman Coulter | | |

## Data collection

Peripheral blood samples were obtained from individuals suspected of having prostate cancer that attended the Urology Clinic at Leicester General Hospital (Leicester, UK) between 24th October 2012 and 15th August 2014. Only patients who had provided informed consent and met the criteria of being biopsy naive, a benign feeling Digital Rectal Examination (DRE) with a PSA level of < 20 ng ml$^{-1}$ and agreeing to undergo a simultaneous 12 core TRUS biopsy and a 36 core transperineal template prostate biopsy (TPTPB) were included in the study. Further details regarding the TPTPB technique are provided in *Nafie et al., 2014b*. A total of 71 males (30 patients diagnosed with benign disease and 41 patients diagnosed with cancer, as confirmed by pathological examination of TPTPB biopsies) met the criteria. Of the 30 patients diagnosed with benign disease; 9 patients were diagnosed with High Grade Prostatic Intraepithelial Neoplasia (PIN), 10 patients were diagnosed with Atypia and 2 patients were diagnosed with Atypical Small Acinar Proliferation. The remainder were diagnosed with benign disease. Of the men diagnosed with prostate cancer, 16 had Gleason 6 disease, 23 had Gleason 7 disease and 2 had Gleason 9 disease on biopsy-based evidence. The clinical features of individuals with benign disease and patients with prostate cancer are provided in *Table 9*.

## The cancer patients dataset used for building the risk prediction model

Data derived from the 41 individuals with prostate cancer were extracted from the dataset shown in *Table 9*. All 41 patients had PSA < 20 ng ml$^{-1}$. However, three of the 41 patients who had a High D'Amico risk were removed because their clinical profiles were very different from those of other high risk patients. They were patients with either a Gleason score 3+3 or had a benign biopsy. In the future, we aim to collect more data from such infrequent patient groups to train the algorithms on patients with such clinical profiles. The remaining 38 patients had PSA levels < 20 ng ml$^{-1}$ and belonged to the D'Amico L/I risk group.

Data were collected from an additional 16 patients with prostate cancer who were diagnosed as having a D'Amico High risk profile (see *Table 10*). Thus, the new cancer patient dataset comprised 54 patients with prostate cancer, of which 38 patients belonged to the D'Amico L/I risk group and all had PSA<20 ng ml$^{-1}$, and 16 patients belong to the D'Amico H risk group and have PSA 4.3 ng

**Table 9.** Patient clinical features.

| Patient group | Gleason score | Number of patients | Age range (years) | PSA range (ng/ml) |
|---|---|---|---|---|
| Benign | Benign | 9 | 64-71 | 5.3–15 |
| Benign | HGPIN | 9 | 54–70 | 5.1–12 |
| Benign | Atypia | 10 | 50–76 | 4.7–19 |
| Benign | ASAP | 2 | 59–60 | 5.3–7.8 |
| Cancer | Gleason 6 | 16 | 55–80 | 4.7–11 |
| Cancer | Gleason 7 | 23 | 53–77 | 4.7–19 |
| Cancer | Gleason 9 | 2 | 65–75 | 6.3–18 |

**Table 10.** Dataset used for differentiating between patients with L/I and H cancer.

| Patient group | Count | % |
|---|---|---|
| L/I | 38 | 70.37 |
| H | 16 | 29.63 |

ml$^{-1}$≤ PSA ≤ 2617 ng ml$^{-1}$. The 16 patients were diagnosed with Gleason scores of: 4+4 = 8 (n = 2), 5+4 = 9 (n = 2), and 4+5 = 9 (n = 11), and 1 patient was diagnosed with small cell cancer. The combined dataset (i.e. 38+16 = 54) comprised 15 patients with Gleason 6 (3+3), 18 patients with Gleason 7 (3+4), 5 patients with Gleason 7 (4+3), 2 patients with Gleason 8 (4+4), 11 patients with Gleason 9 (4+5), 2 patients with Gleason 9 (5+4), and 1 patient with small cell cancer.

Since 11 of those 16 patients had a PSA > 20 ng ml$^{-1}$, their data could only be utilised for building the prostate cancer risk prediction model, as the detection model focuses on detecting prostate cancer in asymptomatic men with PSA< 20 ng ml$^{-1}$.

## Flow cytometric analysis

Peripheral blood (60 ml) was collected from all patients using standard clinical procedures. Aliquots (30 ml) were transferred into two sterile 50 ml polypropylene (Falcon) tubes containing 300 µl of sterilized Sigma Aldrich Lithium Heparin (1000 U/ml, Merck Millipore). Anti-coagulated samples were transferred to the John van Geest Cancer Research Centre at Nottingham Trent University (Nottingham, UK) and processed immediately upon receipt (always within 3 hr of collection). Peripheral blood (60 ml) was mixed with Phosphate Buffered Saline (PBS, 30 ml, Lonza) and layered over Ficoll-Paque (GE Healthcare Life Sciences) in Leucosep tubes (20 ml blood per tube) and then centrifuged at 800 g for 20 min. The peripheral blood mononuclear cell (PBMC) fraction was harvested and washed twice with PBS before being re-suspended in Hyclone fetal bovine serum (FBS, GE Healthcare Life Sciences). Viable cells were counted using trypan blue (0.1 % v/v trypan blue, Santa Cruz) and a haemocytometer. Cells were frozen in 90% v/v FBS, 10% v/v DMSO (Santa Cruz) in aliquots of $10 \times 10^6$ PBMC/vial and stored in liquid nitrogen until phenotypic analysis. At the time of analysis, one vial from each patient was thawed by mixing with 10 ml 'thaw' solution (90% v/v RPMI (Lonza)), 10% v/v CTL wash solution (Cellular Technology Limited) and 10 µl of Novagen Benzonase (Merck Millipore) at room temperature.

PBMCs were centrifuged at 400 g for 5 min followed by resuspension in 1 ml of RPMI (supplemented with 10% v/v FBS, 1% v/v L-glutamine (Lonza)). Cells were rested for 1 hr at 37, after which viable cells were counted using trypan blue dye (Santa Cruz) exclusion. For each monoclonal antibody (mAb) panel shown in *Table 11*, $1 \times 10^6$ cells were washed and incubated in 100 µl of Wash Buffer (PBS +2% w/v Calbiochem bovine serum albumin (BSA, Merck Millipore) +0.02% w/v sodium azide (Sigma)) containing the relevant mAb cocktail for 15 min, after which cells were washed with 1 ml PBS and then incubated in 1 ml LIVE/DEAD Fixable Violet dead stain (Thermo Fisher Scientific) for 30 min. All incubations were performed at 4 protected from light. The cells were washed with PBS and then re-suspended in Beckman Coulter Isoton isotonic buffered saline solution.

Data (on viable cells) were acquired within 1 hr using a 10-color/3-laser Beckman Coulter Gallios flow cytometer and analyzed using Beckman Coulter Kaluza v1.3 data acquisition and analysis software. Controls used a Fluorescence Minus One (FMO) approach. A typical gating strategy for the analyses is presented in *Figure 8*.

## Computational methods

Initially, the GA by *Ludwig and Nunes, 2010* was adopted to identify the best subset of features (i.e. predictors), and thereafter a prediction model was constructed using the Ensemble classifier. This section also explains the metrics adopted for evaluating the performance of the prediction model.

## GA for selecting the best subset of features

The GA is a metaheuristic, commonly used to generate solutions to optimization and search problems. Given the large number of combinations, the process of selecting the best subset of flow

**Table 11.** Antibody panels for measuring the phenotype of Natural Killer cells.

| Antibody | Fluorochrome | Clone no. | Supplier |
| --- | --- | --- | --- |
| Panel 1 | | | |
| DNAM-1 (CD226) | FITC | 11A8 | BioLegend |
| NKG2D (CD314) | PE | 1D11 | eBioscience |
| CD56 | ECD (PE-Texas Red) | N901 | Beckman Coulter |
| CD16 | PerCP-Cy5.5 | 3G8 | BioLegend |
| NKp46 (CD335) | PE-Cy7 | 9E2 | BioLegend |
| NKp30 (CD337) | Alexa Fluor 647 | P30-15 | BioLegend |
| CD3 | Alexa Fluor 700 | UCHT1 | BioLegend |
| CD19 | Alexa Fluor 700 | HIB19 | BioLegend |
| CD8 | APC-Cy7 | SK1 | BioLegend |
| Live/Dead | Dye (violet) | | Thermo Fisher Scientific |
| Panel 2 | | | |
| CD85j (ILT2) | FITC | GHI/75 | Miltenyi Biotec |
| LAIR-1 (CD305) | PE | DX26 | BD Biosciences |
| CD56 | ECD (PE-Texas Red) | N901 | Beckman Coulter |
| CD16 | PerCP-Cy5.5 | 3G8 | BioLegend |
| NKG2A (CD159a) | PC7 (PE-Cy7) | Z199 | Beckman Coulter |
| NKp44 (CD336) | Alexa Fluor 647 | P44-8 | BioLegend |
| CD3 | Alexa Fluor 700 | UCHT1 | BioLegend |
| CD19 | Alexa Fluor 700 | HIB19 | BioLegend |
| CD8 | APC-Cy7 | SK1 | BioLegend |
| LIVE/DEAD | Dye (violet) | | Thermo Fisher Scientific |
| Panel 3 | | | |
| 2B4 (CD244.2) | FITC | C1.7 | BioLegend |
| CD56 | ECD (PE-Texas Red) | N901 | Beckman Coulter |
| CD16 | PerCp-Cy5.5 | 3G8 | BioLegend |
| CD3 | Alexa Fluor 700 | UCHT1 | BioLegend |
| CD19 | Alexa Fluor 700 | HIB19 | BioLegend |
| CD8 | APC-Cy7 | SK1 | BioLegend |
| LIVE/DEAD | Dye (violet) | | Thermo Fisher Scientific |

cytometry features for creating the prediction algorithm is performed using a GA. The GA adopted in the experiments was developed by *Ludwig and Nunes, 2010*. The particular GA performs combinatorial optimization to identify a subset of features that comprises the optimum feature set, in which the order of features has no relation with their importance. The algorithm works by maximising the mutual information between the target y (where y can have a value 1 for cancer or 0 for benign) and the input features (i.e. these are the 32 features listed in *Table 1*). Mutual information is the measure of the mutual dependence between the two variables, i.e. an input feature and the target. Adopting a GA eliminates the computational effort which is necessary to evaluate all the possible combinations of features. The fitness function of the GA (*Ludwig and Nunes, 2010*) is based on the principle of max-relevance and min-redundancy (mRMR), for which the objective is that the outputs of the selected features present discriminant power, thereby avoiding redundancy. The principle of max-relevance and min-redundancy corresponds to searching the set of feature indexes that are mutually exclusive and correlated to the target output. Let $m \times n$ be a feature-by-patient matrix, $X = [x_{ij}]$ with $m$ features and $n$ patients. Thus, the matrix element $x_{ij}$ is the flow cytometry value $i$ of patient $j$. Let $y$ be a vector of size $1 \times n$ which holds the diagnosis of each patient (1 for cancer and

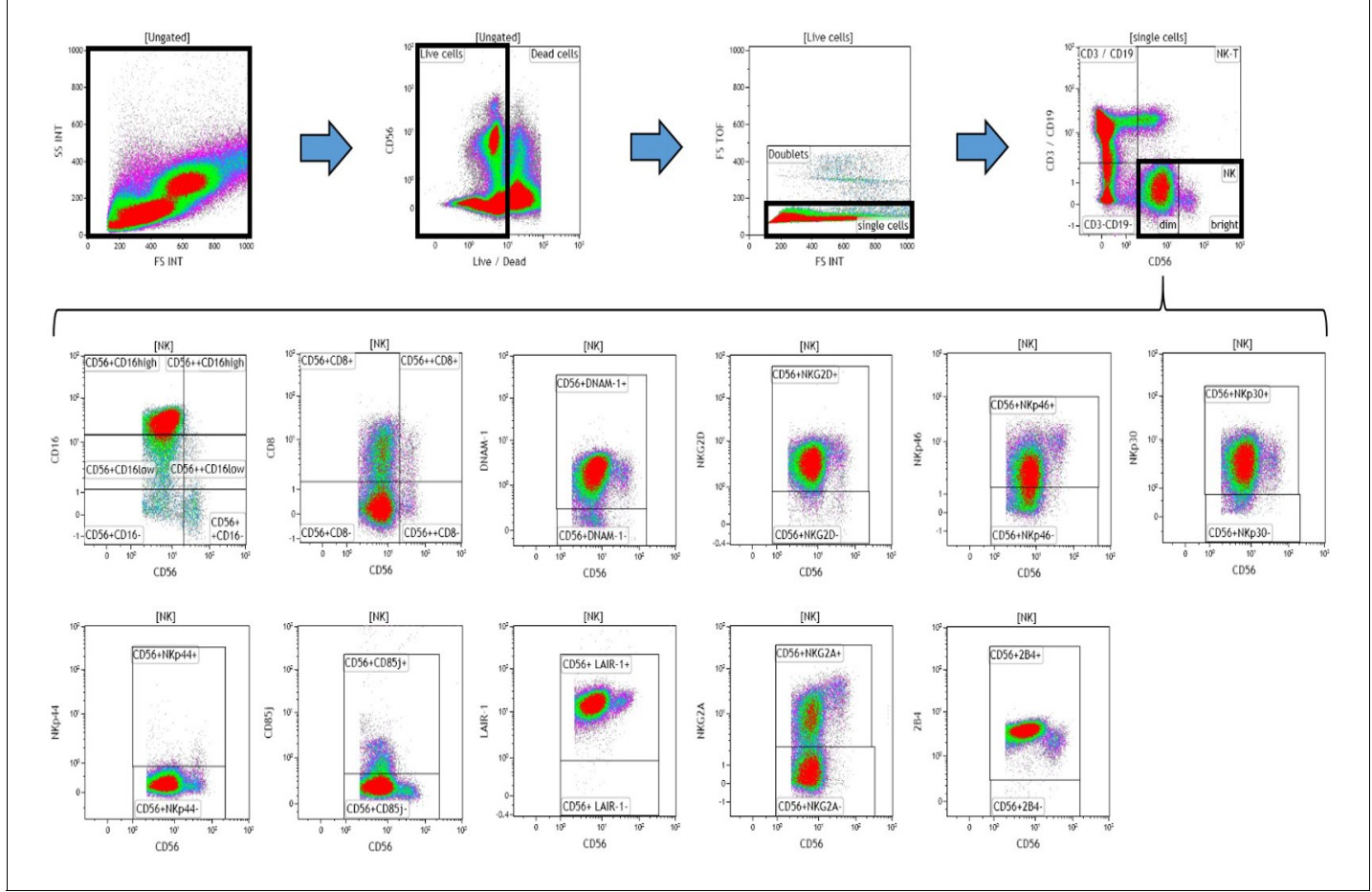

**Figure 8.** Representative gating strategy for analyzing the expression of activating and inhibitory receptors on peripheral blood natural killer (NK) cells. Using density plots, the NK cell phenotypic profiles were determined by first gating on 'live cells' in the forward scatter (FSc) linear vs side scatter (SSc) linear density plot and then gating on single cells (determined by FSc Linear vs FS time of flight). The expression of activating and inhibitory receptors was determined by gating on $CD3^-CD19^-CD56^+$ cells using fluorescence minus one (FMO) controls. The expression of each NK cell receptor was measured using the 'Logical' setting.

0 for benign). Hence, each patient $x$ is mapped to a diagnosis $y$. The GA takes three inputs: 1) the feature-by-patient matrix X; 2) the vector $y$ which holds the corresponding labels for each patient record; and 3) the desired number of features, $\lambda$. The GA returns the IDs of the best subset of features, where the subset has size $\lambda$. GAs stochastically select multiple features from the current population and thus each run of the GA can return different results. Consequently, we proposed an approach to identify the best subset of features by running the algorithm several times and then obtaining the frequency of the subsets.

## Proposed ensemble learning classifier for identifying the presence of prostate cancer

This section discusses the machine learning classifier which was developed for the task of identifying the presence of benign prostate disease or prostate cancer using the identified subset of phenotypic features. The challenging task is that a suitable and reliable classifier must be developed using only 72 patient records. A limitation is that classifiers that have been trained on small sample size data are likely to be unstable because small changes in the training set cause large changes in the classifier. It was for this reason that the Ensemble machine learning classifier was preferred as an approach for developing a more stable and reliable classifier. Ensemble classifiers achieve stability and reliability by constructing many 'weak' classifiers instead of a single classifier and then combine the weak classifiers (i.e. weak learners) to create a more powerful decision rule than that constructed

when using a single classifier. In clinical applications, it is important to construct prediction models which have a low bias, meaning that the classifier suggests fewer assumptions about the form of the target function. Because Ensemble learning makes fewer assumptions about the form of the target function, it was considered to be a suitable classifier for the task. Several techniques for combining the classifiers of an Ensemble model exist and these include Boosting, Bagging, and Random Subspace Dimension.

In the proposed method, the Random Subspace Dimension approach was utilised as a strategy for combining the kNN classifiers, to create the Ensemble of kNN classifiers. In machine learning, the Random Subspace Method (*Ho, 1998*), also called attribute bagging (*Bryll et al., 2003*) or feature bagging, is an Ensemble learning method which attempts to reduce the correlation between estimators in an Ensemble by training them on random samples of features instead of the entire feature set. In the Random Subspace method, classifiers are constructed in random subspaces of the data feature space. These classifiers were combined by simple majority voting in the final decision rule, and we used the k Nearest Neighbor method (see *Figure 9*). In particular, we used the Random Subspace ensemble-aggregation method coupled with k Nearest Neighbours weak learners to produce an Ensemble of classifiers, and this resulted to a better classification rule. Thus, the Random Space modifies the training data set, builds classifiers on these modified training sets, and then combines them into a final decision rule by simple or weighted majority voting.

*Figure 9* provides an overview of the architecture of the proposed kNN Ensemble learning, and the description that follows explains the architecture in more detail. Let $m$ be the number of dimensions (variables) to sample in each learner minus 1. Let $d$ be the number of dimensions in the data, which is the number of predictors in the data matrix X. Let $n$ be the number of learners in the ensemble. The basic random subspace algorithm performs the following steps using the above-mentioned parameters:

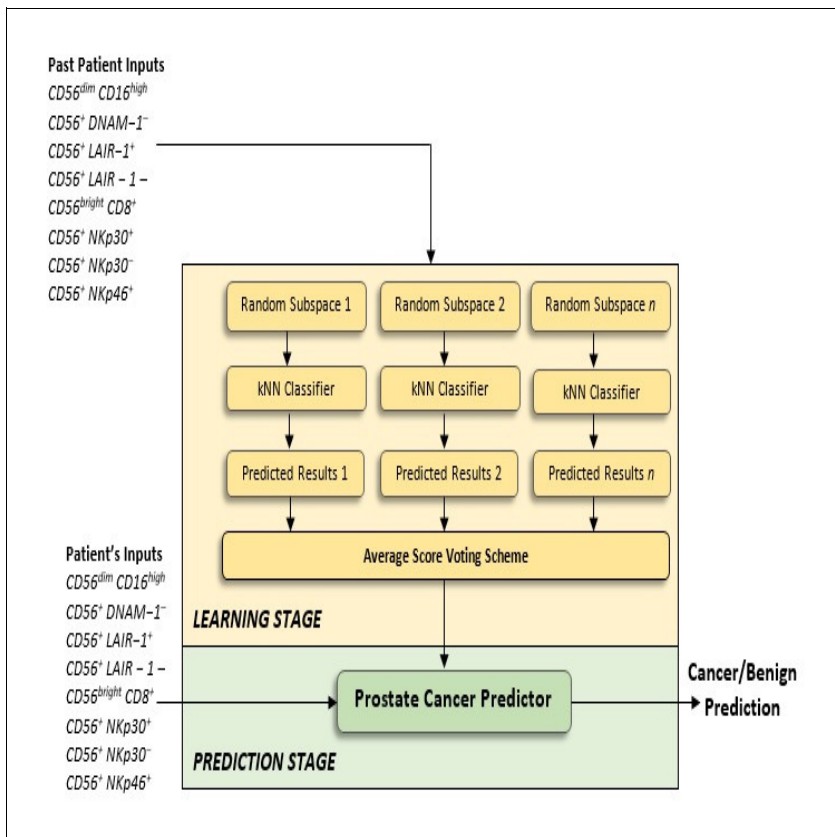

**Figure 9.** Proposed Ensemble Subspace kNN model. Ensembles combine predictions from different models to generate a final prediction. Because Ensemble approaches combine baseline predictions, they perform at least as well as the best baseline model.

1. Choose without replacement a random set of $m$ predictors from the $d$ possible values.
2. Train a weak learner using just the $m$ chosen predictors.
3. Repeat steps 1 and 2 until there are $n$ weak learners.
4. Predict by taking an average of the score prediction of the weak learners and classify the category with the highest average score.

## Performance evaluation measures

A variety of relevant evaluation metrics were adopted for the task of evaluating the performance of the machine learning prostate cancer presence and risk prediction models.

Prostate cancer presence prediction models: Let $|TP|$ be the total number of patients with cancer who were correctly classified as having cancer; $|TN|$ be total the number of individuals with benign disease who were correctly classified as having benign disease; $|FP|$ be the total number of individuals with benign disease who were incorrectly classified as having cancer; $|FN|$ be the total number of patients with cancer who were incorrectly classified as having benign disease; $|P|$ be the total number of patients with cancer that exist in the dataset, where $|P| = |TP| + |FN|$; and $|N|$ be the total number of individuals with benign disease that exist in the dataset, where $|N| = |FP| + |TN|$. The following commonly used evaluation measures can be defined.

$$Accuracy = \frac{|TP| + |TN|}{|TP| + |FP| + |FN| + |TN|}, \in [0, 1].$$ (2)

$$TPR = \frac{|TP|}{|TP| + |FN|}, \in [0, 1].$$ (3)

$$TNR = \frac{|TN|}{|TN| + |FP|}, \in [0, 1].$$ (4)

$$FNR = \frac{|FN|}{|TP| + |FN|} = 1 - Sensitivity, \in [0, 1].$$ (5)

$$FPR = \frac{|FP|}{|FP| + |TN|} = 1 - Specificity, \in [0, 1].$$ (6)

The closer the values of Accuracy, True Positive Rate (i.e. TPR, Sensitivity) and True Negative Rate (i.e. TNR, Specificity) are to 1.0, then the better the classification performance of a system.

The Receiver Operating Characteristic (ROC) is an effective measure for evaluating the quality of a prediction model's performance. The ROC curve has an optimal ROC point which comprises two values: the False Positive Rate (FPR) and the True Positive Rate (TPR) values. The optimal ROC point is computed by function (*Equation 7*) for finding the slope, $S$.

$$S = \frac{Cost(P|N) - Cost(N|N)}{Cost(N|P) - Cost(P|P)} \times \frac{N}{P},$$ (7)

where $Cost(N|P)$ is the cost of misclassifying a positive class (i.e. cancer) as a negative class (i.e. benign); $Cost(P|N)$ is the cost of misclassifying a negative class, as a positive class; $P$, and $N$, are the total instance counts in the cancer and benign class, respectively. The optimal ROC point is identified by moving the straight line with slope $S$ from the upper left corner of the ROC plot ($FPR = 0$, $TPR = 1$) down and to the right, until it intersects the ROC curve.

The Area Under the ROC Curve (AUC) is another important performance evaluation metric which reflects the capacity of a model capacity to discriminate between the data obtained from individuals with benign disease and patients with cancer. The larger the AUC, the better the overall capacity of the classification system to correctly identify benign disease and cancer.

Prostate cancer risk prediction models: When applying the above-mentioned measures to evaluate the performance of the risk prediction models, the Positive class, P, was changed to be the High-risk group and the Negative class, N, was changed to be the L/I group.

## Acknowledgements

The authors acknowledge the financial support of the John and Lucille van Geest Foundation, the Healthcare and Bioscience iNet, an ERDF funded initiative managed by Medilink East Midlands, PROSTaid, and Nottingham Trent University. This work was also supported by a Nottingham Trent University Vice Chancellor PhD Studentship Bursary to SPH. Dr Cosma acknowledges the financial support of The Leverhulme Trust (Research Project Grant RPG-2016–252). The funders had no role in study design, data collection, and analysis, decision to publish, or preparation of the manuscript.

## Additional information

### Competing interests

Georgina Cosma, A Graham Pockley: Named inventor on filed patent application entitled 'Machine learning models and methods for detecting presence and clinical significance of prostate cancer' (Application Number GB1910689.7). The other authors declare that no competing interests exist.

### Funding

| Funder | Grant reference number | Author |
|---|---|---|
| The John and Lucille van Geest Foundation | Core / Programme Grant | A Graham Pockley |
| ERDF | Healthcare and Bioscience iNet Research Grant | A Graham Pockley |
| PROSTaid Prostate Cancer Charity | Funding Support | Stéphanie E McArdle A Graham Pockley |
| Nottingham Trent University | PhD Studentship | Simon P Hood A Graham Pockley |
| Leverhulme Trust | Research Project Grant RPG-2016-252 | Georgina Cosma |

The funders had no role in study design, data collection and interpretation, or the decision to submit the work for publication.

### Author contributions

Simon P Hood, Gemma A Foulds, Data curation, Formal analysis, Validation, Investigation, Visualization, Methodology, Writing - review and editing; Georgina Cosma, Software, Formal analysis, Validation, Investigation, Visualization, Methodology, Writing - original draft, Writing - review and editing; Catherine Johnson, Stephen Reeder, Investigation, Methodology; Stéphanie E McArdle, Formal analysis, Investigation, Methodology, Writing - review and editing; Masood A Khan, Resources, Data curation, Validation, Investigation, Writing - review and editing; A Graham Pockley, Conceptualization, Supervision, Funding acquisition, Investigation, Methodology, Writing - original draft, Project administration, Writing - review and editing

### Author ORCIDs

Georgina Cosma (iD) https://orcid.org/0000-0002-4663-6907

A Graham Pockley (iD) https://orcid.org/0000-0001-9593-6431

### Ethics

Human subjects: Research Protocols were registered and approved by the National Research Ethics Service (NRES) Committee East Midlands and by the Research and Development Department in the University Hospitals of Leicester NHS Trust. All participants were given information sheets explaining the nature of the study and all provided informed consent. All samples were collected by suitably qualified individuals using standard procedures. Ethical approval for the collection and use of samples from the TPTPB cohort (Project Title: Defining the role of Transperineal Template-guided prostate biopsy) was given by NRES Committee East Midlands- Derby 1 (NREC Reference number: 11/EM/3012; UHL11068).

Decision letter and Author response
Decision letter https://doi.org/10.7554/eLife.50936.sa1
Author response https://doi.org/10.7554/eLife.50936.sa2

## Additional files

### Supplementary files
• Source data 1. Prostate Cancer Dataset.

• Transparent reporting form

### Data availability
A spreadsheet of the immune cell phenotypic data has been provided as Source Data File 1.

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
