## [Decision Letter]

**Acceptance summary:**

As Prostate-Specific Antigen (PSA) testing in prostate cancer diagnostics suffers from high false-positive rates, a pressing unmet medical need presents to develop a sensitive and specific biomarker for prostate cancer diagnosis. The current work presents a flow-cytometry based approach to detect prostate cancer that may prove to be of high clinical value.

**Decision letter after peer review:**

Thank you for sending your article entitled "Identifying prostate cancer using Machine Learning of peripheral blood natural killer cell subset phenotyping data" for peer review at *eLife*. Your article is being evaluated by two peer reviewers, and the evaluation is being overseen by Reviewing Editor Wilbert Zwart and Eduardo Franco as the Senior Editor.

As you can appreciate from the detailed reviewer comments below, several important concerns were raised. The reviewers' unedited critiques are copied below.

Reviewer #1:

The authors present a machine-learning-based classification of 1) benign/prostate cancer patients and 2) high/low-risk patients. The authors used multiple machine learning algorithms (genetic algorithm and ensemble learning) and indicated their methods perform better than PSA-based classification.

Overall, though the key elements of the manuscript are machine learning and statistics, the description of both data and algorithms are mostly insufficient and poorly written. It is unclear why a particular algorithm is chosen, and how they are implemented. In particular, the genetic algorithm is typically an optimization algorithm, for which objective function needs to be defined. The genetic algorithm can be used for other problems (e.g., clustering) depends on how the objective function is designed. However, the authors just mention genetic algorithm as if this is a classification method, without defining an objective function. Also, the Ensemble learning algorithm takes multiple simple classification methods, but the choice of the simple classifier can vary and thus needs to be specified. In general, it is not clear if these complicated algorithms are really necessary, as the authors do not show the performance from the simpler classifier (i.e. weak classifier). Also, the text requires extensive re-writing. The text is jargon-rich (especially those related to machine learning), and often repetitive.

Specific comments:

1) Comparisons of the features between benign / prostate cancer patients are presented in figures and tables, but they are difficult to read. For instance, Figure 2 has two panels separated by the category (benign and cancer), but comparison per feature can be done if the authors simply put them side-by-side per feature in one boxplot. Also, per feature, a p-value from a simple t-test can help readers to understand if the features are different or not. Also, the authors should match the order of row/columns in Figure 4A and B.

2) It is unclear why the Ensemble learning is used. The authors should show if the simple classifier is sufficient to predict the outcome and then show if the performance improved by combining the simple classifiers.

3) It is not at all clear what genetic algorithm does. It completely depends on how objective function is chosen, and it is not defined in the text. Also, it is again unclear how the Genetic algorithm can perform feature selection.

4) The authors removed 3 patients with high-risk category (D'Amico High risk), but I do not understand why this is justifiable.

5) The authors discuss yet another Ensemble approach, namely the Random subspace dimension approach. However, the analysis is inconclusive and thus not clear why the authors brought this up.

6) It is not helping at all to mention the outcome without any data shown.

Reviewer #2:

This study extends their own research (Cosma et al., 2017) and tries to predict prostate cancer based on flow cytometric profiling of blood immune cell subsets. The computational algorithm is very straight forward. They first generate 32 features using flow cytometry. Then the genetic algorithm and statistical test is used for feature selection. The ensemble learning classifier is finally used with 10-fold cross validation for evaluation. Compared with their previous study, the technical difference is that they replace KNN classifier (used in the previous study) with the ensemble random subspace method used in this study. It is claimed that more patients are included in this study, and that peripheral blood NK cell phenotyping data is first-time used with computational modeling. As a computational scientist, I personally think that this study is limited in technical contribution. But I cannot fairly judge the contribution in medical domain. Overall the paper is well written and easily understood.

My major concerns are listed below:

1) Looks like that the authors first run the genetic algorithm and statistical test to select subsets of features using the whole patient data. They then run 10 fold-cross validation with selected features. Considering that the feature selection is performed on the whole data, the prediction performance might be boosted. The typical procedure is that feature selection is performed on training set during each folder of cross-validation. The selected features are then applied on the corresponding testing set. Under the current evaluation process, the authors are suggested to include more patients for independent testing. The newly testing patients should not be used for feature selection.

2) It is not justified or explained why all features are used to predict L/I risk caner vs H risk cancer on 54 patients, without using feature selection?

3) It is suggested to provide quantitative comparisons with their own study (Cosma et al., 2017) during the experiments. Thus it would be more technical convincing that the new method is better than the previous one.

[Editors' note: further revisions were suggested prior to acceptance, as described below.]

Thank you for submitting your revised article "Identifying prostate cancer and its clinical risk using Machine Learning of blood NK cell subset phenotyping data" for consideration by *eLife*. Your article has been reviewed by two peer reviewers, and the evaluation has been overseen by a Reviewing Editor and Eduardo Franco as the Senior Editor. The following individuals involved in review of your submission have agreed to reveal their identity: Yongsoo Kim (Reviewer #1); Hongming Xu (Reviewer #2).

The reviewers have discussed the reviews with one another and the Reviewing Editor has drafted this decision to help you prepare another revised submission. We are happy to see the effort you and your co-authors made addressing the first round of reviews. However, our conclusion sis that additional work is needed.

As the editors have judged that your manuscript is of interest, but as described below that additional analyses are required before it is published, we would like to draw your attention to changes in our revision policy that we have made in response to COVID-19 (https://elifesciences.org/articles/57162). First, because many researchers have temporarily lost access to the labs, we will give authors as much time as they need to submit revised manuscripts. We are also offering, if you choose, to post the manuscript to bioRxiv (if it is not already there) along with this decision letter and a formal designation that the manuscript is "in revision at *eLife*". Please let us know if you would like to pursue this option. (If your work is more suitable for medRxiv, you will need to post the preprint yourself, as the mechanisms for us to do so are still in development.)

Overall, the authors revised the manuscript and there is a significant improvement in clarity of technical discussion. In particular, the benefit of the ensemble k-nearest neighbor classifier is well-described in the new version of the manuscript, by comparing it with the weak classifier. However, both reviewers and the reviewing editor felt that a number of issues are not sufficiently addressed at this point, which would be required for the paper to be considered eligible for publication.

Essential revisions:

1).The authors mention the co-linearity issue and found a substantial proportion of features are correlated to each other (376 out of 496 feature pairs; also indicated in Figure 3) in Results paragraph two. The authors claim that the final predictor should not combine features with a high correlation. Are the features sets suggested by the authors (8 features from STAT+GA) indeed not correlated? In Figure 3, we can find a high positive or negative correlation among 15th-18th features. And in statistical analysis, the 14th-18th features are statistically significant (in Table 2), which might correspond to the highly correlated features. If that is the case, it sounds like the authors did something they should not do, according to their own opinion. Please address this.

2) The authors chose λ=4 after examining the stability of GA. I do agree that stability is an important aspect. However, I think it is also important to know how good the performance of the final solution found by GA is. In that regard, it is worth reporting the final mutual information next to the Relative Frequency in Table 3.

3) The authors used the Random subspace dimension approach, which is the best performing. The authors did not present any data to support the claim. This should be provided.

4) The authors claimed that they demonstrated all of 32 features are required. However, the performance of the algorithm needs to be assessed with subsets of the features to make the claim.

5) In the revised version, overall, the authors have tried to address and answer the concerns I listed before. Because of difficulty and time-requirement for collecting more data, only cross-validation was performed in this study. External testing for the presented method is hard for current situation now. So it is suggested to mention this limitation in the paper's discussion part.

[Editors' note: further revisions were suggested prior to acceptance, as described below.]

Thank you for submitting your revised article "Identifying prostate cancer and its clinical risk using Machine Learning of blood NK cell subset phenotyping data" for consideration by *eLife*. Your article has been reviewed by two peer reviewers, and the evaluation has been overseen by a Reviewing Editor and a Senior Editor. The following individuals involved in review of your submission have agreed to reveal their identity: Yongsoo Kim (Reviewer #1); Hongming Xu (Reviewer #2).

We are happy to see the effort you made at amending the paper to accommodate the concerns and suggestions from the reviewers. Once again, we are unable to accept it in its present form for publication. However, we are willing to consider a new revised version if you can address the additional concerns and suggestions below. The reviewers have discussed the reviews with one another and the Reviewing Editor has drafted this decision to help you prepare a revised submission.

The authors have addressed the vast majority of issues raised, and all reviewers are in general satisfied with the revised work. One final question that remained unanswered in the previous round would still require attention at this stage.

Major comments

1) My main concern is still the Revision#4 question, which was asked in the second round review. In the experiments of predicting low/intermediate risk cancer vs high risk cancer, the authors used all 32 features to train the ensemble model by arguing due to the small dataset. But usually if the dataset is small, it is better to use a smaller number of features. In the experiments of benign disease and prostate cancer distinction, the author claimed that 8 features selected by GA+STAT provided the best performance. So it is still suggested to use those 8 features for L/I vs high risk distinction. If using those 8 features cannot provide better performance, the authors can provide some insight discussions about this phenomenon. In addition, for the distinction between L/I and H cancer, 16 more high risk cancer patients were included. Are there any reasons why these 16 patients were not used for cancer and benign distinction?

---

## [Author Response]

As you can appreciate from the detailed reviewer comments below, several important concerns were raised. The reviewers' unedited critiques are copied below.Reviewer #1:The authors present a machine-learning-based classification of 1) benign/prostate cancer patients and 2) high/low-risk patients. The authors used multiple machine learning algorithms (genetic algorithm and ensemble learning) and indicated their methods perform better than PSA-based classification.

Genetic Algorithms (GA) are not machine learning algorithms, they are computational intelligence algorithms (optimisation algorithms), and the output of the GA (selected subset of features) was input into a machine learning algorithm, the Ensemble classifier (these are two separate processes). Please refer to Materials and methods section which explains the algorithms.

The first sentence in Experiment Methodology has been revised in order to point the reader to the section which describes the Genetic Algorithm.

Overall, though the key elements of the manuscript are machine learning and statistics, the description of both data and algorithms are mostly insufficient and poorly written. It is unclear why a particular algorithm is chosen, and how they are implemented. In particular, the genetic algorithm is typically an optimization algorithm, for which objective function needs to be defined. The genetic algorithm can be used for other problems (e.g., clustering) depends on how the objective function is designed. However, the authors just mention genetic algorithm as if this is a classification method, without defining an objective function.

The Genetic Algorithm (GA) performs feature selection and not classification. The section entitled “Genetic Algorithm for Selecting the Best Subset of Features” explains that the GA was implemented for feature selection **–** nowhere in the paper is it stated that the GA was implemented for classification. The features selected by the GA are used to build an Ensemble classifier, as described in the Section “Proposed Ensemble Learning Classifier for the task of Predicting Prostate Cancer”.

Please be more specific with respect to those elements of the manuscript that are considered to be “insufficient” and “poorly written” so that we can specifically address this issue.

Also, the Ensemble learning algorithm takes multiple simple classification methods, but the choice of the simple classifier can vary and thus needs to be specified. In general, it is not clear if these complicated algorithms are really necessary, as the authors do not show the performance from the simpler classifier (i.e. weak classifier).

The algorithms used in the study are necessary and are, in our opinion and practice, relatively uncomplicated. The GA is necessary for finding the best subset of biomarkers because, as described in the “statistical analysis” section, it is not possible to find the best subset of biomarkers which, together as a combination (“feature set”), would make a good predictor of prostate cancer presence using conventional statistical approaches (see Table 6). The best classifier for the data was an Ensemble classifier which was built using the subset of features which were strategically selected using a novel methodology, as described in the Results section “Identifying Prostate Cancer using PSA and Immunophenotyping Data” (Table 5 for results).

Please refer to our response to your Specific Comment (2) where we carried out new experiments to demonstrate that the simple kNN model is not performing as well as the proposed Ensemble model. Please note that we have also carried out experiments with Naïve Bayes, Support Vector Machines, and many other machine learning algorithms (with various settings), but the Ensemble approach performed the best. The paper is already lengthy, and we would like to focus it on discussing the results using the best method. However, to satisfy reviewer 1’s comment we included a section which compares the simple kNN against the more complex approach which is an Ensemble of kNN learners (please refer to our answer to your specific comment (2)). We do not feel that it is relevant to include detailed comparisons of various machine learning approaches (which we can provide if required) as we do not consider that this will add to the findings of the study, but are likely to detract from its key findings and impact.

Also, the text requires extensive re-writing. The text is jargon-rich (especially those related to machine learning), and often repetitive.

We are sorry to hear this. These comments contrast with those of reviewer 2 who stated that “overall the paper is well written and easily understood”. However, we have taken this comment on board and sought input and feedback on presentation and understanding from colleagues who are either experts on machine learning or immunology.

We appreciate the reviewer’s comment and have added additional text to improve understanding for the readership which is not familiar with machine learning. For example, at the end of subsection “A comparison of the best prediction models over 30 runs” we added extra text to explain in lay terms “Discussion on importance of findings.”

Specific comments:1) Comparisons of the features between benign / prostate cancer patients are presented in figures and tables, but they are difficult to read. For instance, Figure 2 has two panels separated by the category (benign and cancer), but comparison per feature can be done if the authors simply put them side-by-side per feature in one boxplot.

Figure 2 has now been revised as requested.

Also, per feature, a p-value from a simple t-test can help readers to understand if the features are different or not.

The data were not normally distributed, as a consequence of which the parametric t-test cannot be used. We therefore used the Kruskal-WallisHtest(also called the "one-way ANOVA on ranks", a rank-based non-parametrictest)which determines if there are any statistically significant differences between two or more groups of an independent variable on a continuous or ordinal dependent variable. We used this to determine if there were any statistically significant differences between the measured NK cell parameters in individuals with benign disease and patients with prostate cancer. Table 4 shows the results of the Kruskal-Wallis test. As indicated in the manuscript, such statistical analysis only identified 4 features which could be potential phenotypic “fingerprints” for distinguishing between the different clinical settings. However, when these 4 features where used to build a predictive model, performance was not satisfactory (see Table 6).

We have extended the sentence on Kruskall-Wallis in the Statistical Analysis section to include justification for using the Kruskal-Wallis test over the parametric alternatives:

“Kruskal-Wallis (also called the "one-way ANOVA on ranks", a rank-based non-parametric test) tests were utilised to check for statistically significant differences between the mean ranks of the NK cell phenotypic features in individuals with benign disease and patients with prostate cancer. As the data were not normally distributed, the non-parametric Kruskal-Wallis test was used as a suitable alternative to its parametric equivalent of one-way analysis of variance (ANOVA).”

Also, the authors should match the order of row/columns in Figure 4A and B.

The Spy plot was removed and Figure 5 was increased in size to make it more readable. The Spy was not needed as the correlated features are now clearly shown in Figure 4 (previously Figure 5.)

2) It is unclear why the Ensemble learning is used. The authors should show if the simple classifier is sufficient to predict the outcome and then show if the performance improved by combining the simple classifiers.

In the Section “Identifying Prostate Cancer using PSA and Immunophenotyping Data” the following paragraph and Table were added. These experiments compare the performance of the simple kNN vs the proposed Ensemble approach, and we hope this is enough evidence to justify the Ensemble approach:

“Finally, the experiments discussed in this Section thus far utilised a machine learning model comprised of an Ensemble of kNN learners (see Section “Proposed Ensemble Learning Classifier for the task of Predicting Prostate Cancer”). Before ending the discussions in this Section, the results of experiments carried out to determine the impact of using the proposed Ensemble method over the simple kNN classifier are summarised. Table 7 shows the performance of a simple kNN tuned when setting the value of k nearest neighbours to 2, 5, and 10 and the distance metric the Euclidean. The last column of Table 7 shows the difference in performance of the two methods. The proposed method, denoted as EkNN, returned better performance than all other kNN alternatives, and hence resulted in higher Mean Accuracy values (+) and lower Standard Deviation values (Std.). Lower Standard Deviation values are an indicator of a more stable and reliable model, since the average values are clustered closely around the mean.”

3) It is not at all clear what genetic algorithm does. It completely depends on how objective function is chosen, and it is not defined in the text. Also, it is again unclear how the Genetic algorithm can perform feature selection.

We respectfully refer the reviewer to section Results – Experiment Methodology, which explains how the GA was used. An extensive explanation is also presented in Materials and methods – Genetic Algorithm for Selecting the Best Subset of Features which also explains the objective function used **“**The fitness function of the Genetic Algorithm is based on the principle of max-relevance and min-redundancy (this is the well-known mRMR), for which the objective is that the outputs of the selected features present discriminant power, thereby avoiding redundancy. The principle of max-relevance and min-redundancy corresponds to searching the set of feature indexes that are mutually exclusive and totally correlated to the target output”. More details about the GA and the objective function can be found in Ludwig and Nunes, 2010 (to which reference is made in the manuscript).

We have modified section “Identifying Predictors from Immunophenotyping Data using a Genetic Algorithm” and the first sentence to improve clarity of the aim of the Genetic Algorithm. It previously said “the aim of this method”, and this may have been the cause of confusion. We apologise and hope that this clarifies the aim of using a GA.

Revised text as follows: “The aim of the Genetic Algorithm is to identify a subset of features which, as a combination, provide an NK cell-based immunophenotypic “fingerprint” which can identify if an asymptomatic individual with PSA levels below 20 ng ml^-1^ has benign disease or prostate cancer in the absence of definitive biopsy-based evidence.”

We respectfully refer the reviewer to section Results – Experiment Methodology, which explains how the GA was used. An extensive explanation is also presented in Materials and methods – Genetic Algorithm for Selecting the Best Subset of Features which also explains the objective function used **“**The fitness function of the Genetic Algorithm is based on the principle of max-relevance and min-redundancy [this is the well-known mRMR], for which the objective is that the outputs of the selected features present discriminant power, thereby avoiding redundancy. The principle of max-relevance and min-redundancy corresponds to searching the set of feature indexes that are mutually exclusive and totally correlated to the target output”. More details about the GA and the objective function can be found in Ludwig and Nunes, 2010 (to which reference is made in the manuscript).

4) The authors removed 3 patients with high-risk category (D'Amico High risk), but I do not understand why this is justifiable.

We thank the reviewer for highlighting this issue, which does indeed require clarification. On inspection, the clinical profiles of the three patients with low PSA <20 ng/ml and high D'Amico risk were different to the rest of the high risk patients, and were removed since more examples from patients with similar profiles to those three would need to be added to the dataset for the machine learning algorithm to be able to robustly learn those (more complex) profiles.

We have provided a clearer explanation as to the rationale/reason for excluding data from these 3 patients from the analysis in the revised manuscript. The following justification as to why the three patients were removed has been added in the manuscript in subsection “The cancer patients dataset”:

“However, three of the 41 patients who had a High D'Amico risk were removed because their clinical profile was very different to that of the other high risk patients. They were patients with either a Gleason score 3+3 or had a benign biopsy. In the future we aim to collect more data from such infrequent patient groups in order to train the algorithms on patients with such clinical profiles.”

5) The authors discuss yet another Ensemble approach, namely the Random subspace dimension approach. However, the analysis is inconclusive and thus not clear why the authors brought this up.

The Materials and methods Section “Proposed Ensemble Learning Classifier for the task of Predicting Prostate Cancer” discusses the methods used. In the paper we only discuss and use one method, namely the Ensemble Random Space Approach which used an ensemble of kNN learners. We would like to stress that these are not new / additional methods, rather these are the methods that were described and used in the Results section (and therefore the same one). The ensemble classifier we refer to is also called the Ensemble Random Subspace Machine Learning classifier, as explained in the section mentioned above. *eLife* requires that the Materials and methods section is placed after Results.

6) It is not helping at all to mention the outcome without any data shown.

It is not clear what the reviewer means here. However, we hope that addressing the reviewer’s previous comments has addressed this comment.

Reviewer #2:This study extends their own research (Cosma et al., 2017) and tries to predict prostate cancer based on flow cytometric profiling of blood immune cell subsets. The computational algorithm is very straight forward. They first generate 32 features using flow cytometry. Then the genetic algorithm and statistical test is used for feature selection. The ensemble learning classifier is finally used with 10-fold cross validation for evaluation. Compared with their previous study, the technical difference is that they replace KNN classifier (used in the previous study) with the ensemble random subspace method used in this study. It is claimed that more patients are included in this study, and that peripheral blood NK cell phenotyping data is first-time used with computational modeling. As a computational scientist, I personally think that this study is limited in technical contribution.

We thank the reviewer for these positive comments – they have understood the paper perfectly. We would like to stress that the contribution to the medical domain is significant, as the “holy grail” of prostate cancer diagnosis and management is to be able to clearly distinguish benign prostate disease (no cancer) and non-clinically significant prostate cancer (neither of which require treatment) from clinically significant prostate cancer (which requires further investigation and treatment). This is currently not possible using the PSA test alone and without the use of invasive biopsies that are extremely uncomfortable and associated with a high rate (~5%) of significant and potentially life-threatening side-effects. Furthermore, “standard” biopsies only provide a definitive diagnosis in ~30% of cases. It should also be noted that 15% of men with “normal” PSA levels typically have prostate cancer, with 15% of these cancers being high-grade. Given the poor diagnostic specificity of PSA, PSA-based prostate cancer screening is not currently supported by the NHS or promoted in any other country. Reliable diagnosis of prostate cancer based on PSA levels alone is therefore not possible and must be confirmed using approaches such as invasive biopsies (see above) and/or MRI scans. However, it should also be noted that with respect to MRI scans, ~25% of “positive” MRI scans and ~20% of “negative” MRI scans can be incorrect.

Asymptomatic men with higher than normal PSA levels, but less than 20ng/ml pose significant problems to the clinician because although only 30%-40% of these men will have prostate cancer, all currently must undergo potentially unnecessary invasive prostate biopsies. It is therefore essential to develop better approaches for distinguishing benign disease and low-risk/grade or small volume intermediate-risk prostate cancer which very rarely require treatment from clinically-significant disease which is likely to progress and requires treatment. Distinguishing men with prostate cancer which is unlikely to progress (for whom “active surveillance” is the most appropriate approach), from men with prostate cancer which is likely to progress and requires treatment is a significant clinical challenge and unmet clinical need. Inappropriate assignment of men to potentially life-threatening invasive procedures and lifelong surveillance for prostate cancer has significant psychological, quality of life, financial and societal consequences.

We expect that our approach to accurately determine the presence of prostate cancer and its clinical significance will avoid the need for up to 70% of prostate biopsies, thereby sparing a significant number of men with benign disease or low risk cancer from unnecessary invasive biopsies and other procedures and also reduce demands of providing healthcare and treatment costs.

But I cannot fairly judge the contribution in medical domain.

The following paragraph has been added to the “Potential Impact” section as a summary for the lay reader:

“Currently available screening methods and tests for prostate cancer lack accuracy and reliability, the consequence of which is that many men unnecessarily undergo invasive tests such as biopsy and/or are misdiagnosed as having the disease. […] If prostate cancer is detected, the second part of the tool will detect the clinical risk of the disease (low, intermediate, high) which will help the clinician decide whether the patient requires no further investigation/treatment (“watch and wait”) or whether further investigation and treatment are required.”

Overall the paper is well written and easily understood.My major concerns are listed below:1) Looks like that the authors first run the genetic algorithm and statistical test to select subsets of features using the whole patient data. They then run 10 fold-cross validation with selected features. Considering that the feature selection is performed on the whole data, the prediction performance might be boosted. The typical procedure is that feature selection is performed on training set during each folder of cross-validation. The selected features are then applied on the corresponding testing set. Under the current evaluation process, the authors are suggested to include more patients for independent testing. The newly testing patients should not be used for feature selection.

The features were indeed selected by applying a GA on the entire set (and without using the classifier). However, we used the random nature of the GA to our advantage. Given that the GAs returned different solutions in each iteration, we devised a new methodology for selecting the features by running the GA 30 times to find the most frequent and stable subset of features (Table 5). By doing this, we were confident that we had chosen the best feature set. With this limitation in mind, it is important to mention that the selected features have been considered by expert immunologists who are all co-authors of the paper in order to ensure that the selected features “made sense” from an immunological and clinical perspective. We hope that the reviewer accepts our response. Although it would be straightforward to carry out experiments in which we leave out a subset of the cases for testing the features selected, the dataset is comparatively small and we would be less confident in the results compared to those derived using the methodology described in the paper.

Although the dataset was complex, it was small in comparison to gene array detests that are typically analysed, and the use of the approaches described herein to generate meaningful clinical information from the dataset which we have is a unique element of the study and contribution to knowledge and the literature.

It should also be noted that the dataset on which the manuscript is based is novel, unique and “one of its kind” in the world. The samples were collected by Co-Author Professor Khan (a clinical urologist at the University Hospitals of Leicester) and, as indicated above, the contribution to the medical domain is significant, as the “holy grail” of prostate cancer diagnosis and management is the clear distinguishing benign disease and non-clinically-significant prostate cancer (neither of which needs treatment) from clinically-significant prostate cancer (which requires further investigation and treatment). This is currently not possible based on the PSA test alone and without the use of invasive biopsies that are extremely uncomfortable and associated with a high rate (~5%) of significant and potentially life-threatening side-effects. Inappropriate assignment of men to potentially life-threatening invasive procedures and lifelong surveillance for prostate cancer has significant psychological, quality of life, financial and societal consequences.

2) It is not justified or explained why all features are used to predict L/I risk caner vs H risk cancer on 54 patients, without using feature selection?

We have added the following text in the revised manuscript to clarify this issue:

“Although the combination of 8 biomarkers defined in our previous study discussed in subsection “Identifying Prostate Cancer using PSA and Immunophenotyping Data” was suitable for detecting the presence of cancer, a second, but key clinical question relates to the clinical significance of any prostate cancer which is present. The work described in this subsection, was performed subsequent to our first study and revealed that all 32 phenotypic features are required to distinguish between low/intermediate risk cancer (L/I) and high risk (H) cancer. However, we expect to be able to identify a subset of these features as the datasets increase and the prediction model is retrained on the larger dataset. As indicated above, the generation and delivery of additional datasets is beyond the scope of this paper.”

3) It is suggested to provide quantitative comparisons with their own study (Cosma et al., 2017) during the experiments. Thus it would be more technical convincing that the new method is better than the previous one.

The Discussion section compares the findings of this study with those of the previous study. We have added an extra bold sentence “Comparing results to the previous study:” to make the explanation more apparent.

[Editors' note: further revisions were suggested prior to acceptance, as described below.]

Essential revisions:1) The authors mention the co-linearity issue and found a substantial proportion of features are correlated to each other (376 out of 496 feature pairs; also indicated in Figure 3) in Results paragraph two. The authors claim that the final predictor should not combine features with a high correlation. Are the features sets suggested by the authors (8 features from STAT+GA) indeed not correlated? In Figure 3, we can find a high positive or negative correlation among 15th-18th features. And in statistical analysis, the 14th-18th features are statistically significant (in Table 2), which might correspond to the highly correlated features. If that is the case, it sounds like the authors did something they should not do, according to their own opinion. Please address this.

The 8 features (Features: 2, 20, 27, 28, 14,15,16,17) which were used to implement the prediction model are not highly positively correlated (i.e. the darkest red). As shown in Figure 3, and the rows corresponding to each feature, the only set of features which are highly correlated are features 27 and 28 (negative correlation). There are no other correlations amongst the selected set of features.

The following paragraph has been added to the manuscript in subsection “Distinguishing Between Benign Prostate Disease and Prostate Cancer: Genetic Algorithm”.

“Referring back to Figure 3 and the correlation values between the selected features 2, 20, 27, 28, 14, 15, 16, 17, it is shown that these features do not have a strong positive correlation. Although there is a strong negative correlation between features 27 and 28, we decided to keep both features since these were selected by the feature selection method.”

The wording of a sentence in subsection “Distinguishing Between Benign Prostate Disease and Prostate Cancer: Statistical Analysis of NK Cell Phenotypic Features and PSA levels” has been rephrased to:

“These difficulties are compounded by the challenge of identifying the best combination of predictors which comprise n number of features, and that features within a combination, ideally, should not correlate.”

Please note: there is no major disadvantage to have correlated features in the feature set (between inputs only) which is used to train the machine learning model. We try to avoid including features which highly correlate because when two features are highly correlated, only one of those features is most likely to be useful, and it will probably be possible to remove one of the features. The main advantage of removing one of the features which are highly correlated is for dimensionality reduction purposes. Our feature set is small and therefore, in this case, there is no harm done to keep both features 27 and 28 until we can evaluate these features further using a larger dataset.

The following explanation has also been added to the paper in subsection “Distinguishing Between Benign Prostate Disease and Prostate Cancer: Statistical Analysis of NK Cell Phenotypic Features and PSA levels”:

“It is important to evaluate correlations between features, because if two features are highly correlated, then only one of these could serve as a candidate predictor. However, there may be occasions were both features are needed and besides the impact of this on the dimensionality of the dataset, there is no other negative impact. Furthermore, when two features are highly correlated and are important, it may be difficult to decide which feature to remove.”

2) The authors chose λ=4 after examining the stability of GA. I do agree that stability is an important aspect. However, I think it is also important to know how good the performance of the final solution found by GA is. In that regard, it is worth reporting the final mutual information next to the Relative Frequency in Table 3.

The concept was to find the most frequent (and hence promising and stable) subset of features over various iterations using an optimisation method in order to speed up the search process. For this reason we utilised the GA proposed by Ludwig and Nunes and “wrapped it around” our experimental methodology to find the best set of features, as described in the paper. As described in Minor Point #1 below: Ludwig’s and Nune’s Feature selector which we utilised performs combinatorial optimisation by using Genetic Algorithms, and it is based on the principle of minimum-redundancy/maximum-relevance (mRMR), which maximizes the mutual information indirectly. The output of the method by Ludwig and Nunes is a vector with the indexes of the features that composes the optimum feature set, in which the order of features has no relation with their importance. Therefore, as MI was not the only method used by the feature selector, it is not appropriate to include MI values in Table 3, as this would suggest that this was the only method used for feature selection, which it was not.

3) The authors used the Random subspace dimension approach, which is the best performing. The authors did not present any data to support the claim. This should be provided.

We have now extended the comparisons to include other machine learning classifiers, each of which were tuned to achieve their highest accuracy for the task.

We have updated section “Comparing the performance of the proposed Ensemble kNN vs a single kNN model”, to include other conventional machine learning classifiers when tuned to achieve their best performance for the task. The results have been included in Table 5 (which is Table 6 in the revised manuscript).

We have retitled the subsection as follows: “Comparing the performance of the proposed Ensemble Subspace kNN classifier with alternative classifiers.”

We have updated the content of the subsection as follows:

“The experiments discussed thus far utilised a machine learning model comprised of an Ensemble of kNN learners (see Section “Proposed Ensemble Learning Classifier for the task of Predicting Prostate Cancer”). […] Naive Bayes was the least efficient classifier, and although it returned the lowest ORP FPR, it also returned the lowest ORP TPR, lowest AUC and Accuracy values; and its Std. values were also higher than those of the EkNN model.”

Sentence “The kNN Ensemble Learning classifier was chosen as being the most suitable for the data and task at hand.” has now been updated to “An Ensemble Subspace kNN classifier was developed for the task at hand.”

4) The authors claimed that they demonstrated all of 32 features are required. However, the performance of the algorithm needs to be assessed with subsets of the features to make the claim.

Although we do see the point the reviewers are making, we would have carried out this analysis if we had a larger dataset. The existing analysis was carried out with a thorough experimental methodology to determine whether there is an underlying pattern that can be detected by the proposed algorithm in predicting whether a cancer patient is in the L/I or H group. Using our proposed model and the full set of features, the proposed model was able to find a pattern and return high performance values during the k-fold and the independent test, and this is a significant finding. However, the dataset is small to be able to confidently identify the most promising subset of predictors which as a combination can be utilised to build a classifier. Therefore, before excluding any features as predictors of L/I or H we wish to explore these with a larger dataset. For this reason, we performed thorough experiments using all features, identified that the proposed machine learning classifier can find a pattern in differentiating between patients with benign and cancer disease, but more experiments will be needed with a larger dataset before we start to exclude features from a set of promising predictors of disease stage. As already described in the paper “Of those 54 patient records, a total of 10 randomly selected records (5 from the L/I group and 5 from the H group) were extracted from the dataset such that they can be used at the testing (mini clinical trial) stage. To ensure thorough experiments, a rigorous methodology was adopted. More specifically, a 10-fold cross validation method was adopted, and the experiments were run in 30 iterations, for which each iteration provided an average test result across 10 folds.”

We noticed some inconsistency in the usage of the words validation and testing with our previous discussion of results, and we have made small updates to the manuscript to improve consistency in terminology.

5) In the revised version, overall, the authors have tried to address and answer the concerns I listed before. Because of difficulty and time-requirement for collecting more data, only cross-validation was performed in this study. External testing for the presented method is hard for current situation now. So it is suggested to mention this limitation in the paper's discussion part.

We have updated the manuscript to address the reviewers comment. The changes are outlined below.

We have added the following paragraph to the last paragraph of subsection “Comparing results to the previous study” where future work is mentioned.

“Future work involves collecting more patient samples to conduct further testing of the proposed machine learning models. In terms of future work from a computational perspective, once we have a larger patient dataset we plan to design deep learning models and compare their performance to the conventional machine learning model which was proposed in this paper.”

[Editors' note: further revisions were suggested prior to acceptance, as described below.]

Major comments1) My main concern is still the Revision#4 question, which was asked in the second round review. In the experiments of predicting low/intermediate risk cancer vs high risk cancer, the authors used all 32 features to train the ensemble model by arguing due to the small dataset. But usually if the dataset is small, it is better to use a smaller number of features. In the experiments of benign disease and prostate cancer distinction, the author claimed that 8 features selected by GA+STAT provided the best performance. So it is still suggested to use those 8 features for L/I vs high risk distinction. If using those 8 features cannot provide better performance, the authors can provide some insight discussions about this phenomenon. In addition, for the distinction between L/I and H cancer, 16 more high risk cancer patients were included.

We thank the reviewers for their comments. The question consists of 2 parts, so we will address those separately.

We have updated the paper by adding the following explanation to address the first comment.

“The dataset that was utilised to identify the biomarker (that comprised 8 features) for detecting the presence of prostate cancer (i.e. benign prostate disease vs prostate cancer) in 71 men, and thus it was large enough to perform the combinatorial feature selection task for finding the best subset of features. […] The combinatorial feature selection task to identify the best subset of features for the risk prediction task will be performed once a larger dataset is available.”

Please note that experiments using the same features that were detected to predict the presence of prostate cancer (i.e. benign prostate disease vs prostate cancer) were not suitable for predicting the risk (L/I vs H) of any prostate cancer that was present. This was expected since the tasks are different, and so it was not appropriate to report those results since the optimisation algorithm was searching for a set of features to differentiate between benign prostate disease and prostate cancer, and not risk (L/I vs H) of any prostate cancer that was present. This explanation was not added to the revised version of the paper as the above text justifies why the biomarker based on 8 features was unsuitable.

Are there any reasons why these 16 patients were not used for cancer and benign distinction?

1. We have updated the title of the section “The cancer patients’ dataset” to “The cancer patient dataset used for building the risk prediction model” to improve clarity.

2. Updated the section “The cancer patient dataset used for building the risk prediction model” to include the following explanation, highlighted in blue text in the paper.

3. The 16 patients were diagnosed with Gleason scores of: 4+4=8 (n=2), 5+4=9 (n=2), and 4+5=9 (n=11), and 1 patient was diagnosed with small cell cancer.

4. Since 11 of those 16 patients had a PSA $>20$ ng ml $^{-1}$, their data could only be utilised for building the prostate cancer risk prediction model, as the detection model focuses on detecting prostate cancer in asymptomatic men with PSA$<20$ ng ml $^{-1}$.